# Adaptive Resolution Residual Networks — Generalizing Across Resolutions Easily and Efficiently

**Léa Demeule**
*lea.demeule@mila.quebec*
*Mila - Quebec AI Institute, Université de Montréal*

**Mahtab Sandhu**
*mahtab.sandhu@mila.quebec*
*Mila - Quebec AI Institute, Université de Montréal*

**Glen Berseth**
*glen.berseth@mila.quebec*
*Mila - Quebec AI Institute, Université de Montréal, and CIFAR*

**Reviewed on OpenReview:** *https://openreview.net/forum?id=kTh5tFd1Mq*

## Abstract

The majority of signal data captured in the real world uses numerous sensors with different resolutions. In practice, most deep learning architectures are *fixed-resolution*; they consider a single resolution at training and inference time. This is convenient to implement but fails to fully take advantage of the diverse signal data that exists. In contrast, other deep learning architectures are *adaptive-resolution*; they directly allow various resolutions to be processed at training and inference time. This provides *computational adaptivity* but either sacrifices *robustness* or *compatibility with mainstream layers*, which hinders their use. In this work, we introduce *Adaptive Resolution Residual Networks* (ARRNs) to surpass this tradeoff. We construct ARRNs from *Laplacian residuals*, which serve as generic *adaptive-resolution* adapters for *fixed-resolution* layers. We use smoothing filters within Laplacian residuals to linearly separate input signals over a series of resolution steps. We can thereby skip Laplacian residuals to cast high-resolution ARRNs into low-resolution ARRNs that are computationally cheaper yet *numerically identical* over low-resolution signals. We guarantee this result when Laplacian residuals are implemented with *perfect* smoothing kernels. We complement this novel component with *Laplacian dropout*, which randomly omits Laplacian residuals during training. This regularizes for robustness to a distribution of lower resolutions. This also regularizes for numerical errors that may occur when Laplacian residuals are implemented with *approximate* smoothing kernels. We provide a solid grounding for the advantageous properties of ARRNs through a theoretical analysis based on *neural operators*, and empirically show that ARRNs embrace the challenge posed by diverse resolutions with *computational adaptivity*, *robustness*, and *compatibility with mainstream layers*.

Efficient problem-solving strategies typically allocate effort according to difficulty. Efficient deep learning architectures may therefore be created by incorporating a form of *computational adaptivity* that is conscientious of the difficulty of individual data points. The case of tasks involving image data, audio data, volumetric data, or any other form of signal data is uniquely positioned to benefit from such an approach. Signals have no universal resolution; there is, instead, a diversity of resolutions that are contingent on the sensors used at the time of capture; there is, therefore, a concrete opportunity for impact given the varying difficulty of individual data points across common data modalities. Signals also conform to a mathematical structure that is well understood, which enables the implementation of inductive biases that can aid in finely quantifying and decomposing this notion of difficulty within an architecture.

To this end, we propose *Adaptive Resolution Residual Networks* (ARRNs), a novel architecture for tasks involving signal data that addresses a gap in the capabilities of prior methods, which either lack *computational adaptivity*, lack *robustness*, or lack *compatibility with mainstream layers*. We begin by outlining a typology of existing methods. We identify three categories: *fixed-resolution*; *adaptive-resolution* through *variable sampling window*; *adaptive-resolution* through *variable sampling density*.

**Fixed-resolution.** In the case of transformer architectures and certain other architectures, resolution is a property that must remain fixed for evaluation to be possible, although signals of different resolutions can be interpolated to the resolution of the architecture to allow evaluation. This solution is sufficient for certain applications. This solution is however incapable of delivering *computational adaptivity* in the sense that it cannot reduce its computational cost at lower resolutions. We thus find this approach unsatisfactory for our purpose.

**Adaptive-resolution through variable sampling window.** In the case of fully convolutional architectures and a range of other architectures, resolution is a property that can be adapted at evaluation. This solution achieves *computational adaptivity* through translation equivariant layers or permutation equivariant layers that can conform to arbitrary resolutions. This solution more formally adapts its resolution by varying its *sampling window* rather than its *sampling density*, which is a critically important consideration for *robustness*. These two means of varying resolution are not equivalent: varying the *sampling window* is analogous to resizing the frame delineating a painting to offer a larger or smaller field of view into the scene; varying the *sampling density* is analogous to changing the quantity and size of brush strokes that make a painting to convey the scene in a finer or coarser degree of detail. We illustrate this distinction in Figure 10 and Figure 11. In the context of natural images, the *sampling window* tends to be consistent, as camera systems have similar optics; the *sampling density* is however subject to vary significantly, as camera systems have sensors that range widely in resolution. This hints at a mismatch between the nature of resolution changes within mainstream architectures and within natural image datasets. In certain applications, it is possible for architectures with an adaptive *sampling window* to learn some degree of invariance to *sampling density*, which can attenuate this mismatch. In Figure 1, however, we demonstrate the adaptive *sampling window* of mainstream architectures can be inappropriate for handling even mild distribution shifts between training resolution and inference resolution in the case of natural images. The training resolution of each architecture is set to be the maximal resolution of each dataset. The inference resolution of each architecture is then swept across a range of lower resolutions, *evaluating directly at that resolution.* Further experimental details are given in section 4. We observe a regression to near-random accuracy by the point inference resolution is halved relative to training resolution, highlighting the lack of *robustness* of this approach.

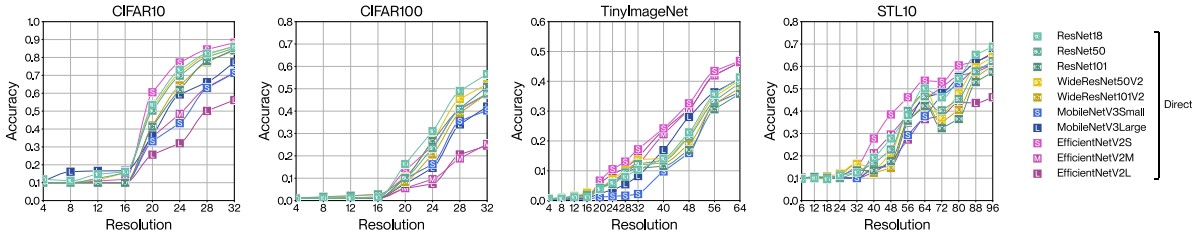

Figure 1: Accuracy of mainstream architectures at various resolutions after training at the full dataset resolution. Evaluation is performed *directly at the inference resolution.*

In Figure 2, we show these architectures display much greater robustness across various resolutions when *evaluating after an interpolation step that ensures the inference resolution matches the training resolution.* This effectively holds constant the *sampling window* while accounting for the change in resolution by varying the *sampling density*, which addresses exactly the discrepancy we have described in the case of natural images. This has the shortcoming of negating any computational benefit that would be gained by performing inference at a lower resolution, as it amounts to treating the architectures as *fixed-resolution*. We cannot find a satisfying solution to our problem through this approach, as it trades away *computational adaptivity* for *robustness*.

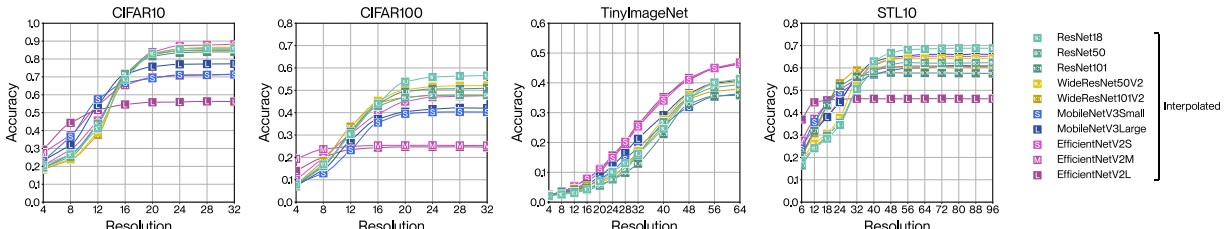

Figure 2: Accuracy of mainstream architectures at various resolutions after training at the full dataset resolution. Evaluation is performed *after interpolation to the training resolution.*

**Adaptive-resolution through variable sampling density.** We wish to obtain both *computational adaptivity* and *robustness*, therefore we must find a way to vary the *sampling density* of an architecture directly. We effectively dedicate our contribution to solving this problem elegantly. We first overview prior methods that achieve both *computational adaptivity* and *robustness*, but which all lack *compatibility with mainstream layers* (section 1). We provide an overview of the fundamental notions of signal processing that allow our work to be formulated and thoroughly define our notation (subsection A.1). We introduce Laplacian pyramids as a stepping stone towards our contribution (section 2). We introduce Laplacian residuals and show that high-resolution ARRNs can be turned into low-resolution ARRNs that are computationally cheaper by simply skipping Laplacian residuals (subsection 3.1). We prove that skipping Laplacian residuals yields *numerically identical* results compared to using all Laplacian residuals when the input signal has a low-resolution and when Laplacian residuals are implemented using *perfect* smoothing kernels. We formulate Laplacian dropout as a training augmentation that randomly omits Laplacian residuals (subsection 3.2). We theoretically motivate its usefulness as a regularizer for low-resolution robustness and as an error correction mechanism for Laplacian residuals implemented using *approximate* smoothing kernels. We perform a set of experiments showing (subsection 4.1) that our method yields stronger robustness at lower resolutions compared to mainstream methods; (subsection 4.2) that our method enables significant computational savings through adaptation; (subsection 4.3) that our method is capable of generalizing across layer types in a way that far surpasses prior *adaptive-resolution* architectures with variable *sampling density*; (subsection 4.4) that our theoretical guarantee for adaptation using *perfect* smoothing kernels holds empirically; (subsection 4.5) that our theoretical interpretation of the dual regularizing effect of Laplacian dropout also holds empirically.

# 1 Related Works

We review related works that allow the formulation of *adaptive-resolution* architectures with *varying sampling density*. We also survey other related works that bear some similarity to Laplacian residuals in section A.2.

**Adaptive-resolution through variable sampling density with neural operators.** We begin our search for means of varying the sampling density of whole architectures by first considering a single convolutional layer. We can see that varying the sampling density of the layer implies changing the sampling density of the kernel itself while preserving the role it plays as a feature extractor. A useful tool to this end is functional analysis; we can think of the kernel as a *discrete function* $k[ \cdot ]_n : \mathbf{X}_n \to \mathbb{R}^{f_{l+1} \times f_l}$ defined over a *discrete coordinate* space $\mathbf{X}_n \subset \mathbb{R}^d$; we can imagine that it that holds a finite number of samples $|\mathbf{X}_n|$ of an underlying *continuous function* $k( \cdot ) : \mathbf{X} \to \mathbb{R}^{f_{l+1} \times f_l}$ defined over a *continuous coordinate* space $\mathbf{X} \subset \mathbb{R}^d$ where $\mathbf{X}_n \subset \mathbf{X}$; we can then make some assumptions about the space of functions we are working with to allow interpolation into an equivalent *discrete function* $k[ \cdot ]_u : \mathbf{X}_u \to \mathbb{R}^{f_{l+1} \times f_l}$ that more finely ($|\mathbf{X}_u| > |\mathbf{X}_n|$) or coarsely ($|\mathbf{X}_u| < |\mathbf{X}_n|$) covers the same continuous coordinate space; we can thus cast the layer from its original sampling density $|\mathbf{X}_n|/\text{area}(\mathbf{X})$ to a new sampling density $|\mathbf{X}_u|/\text{area}(\mathbf{X})$. A next logical step is to conceptualize every layer that composes an architecture as an *operator* that expresses *a map between inputs and outputs that are functions*; we gain the ability to adapt the sampling density of the entire architecture if we ensure every layer can be converted between a *discrete operator form* and a *continuous operator form* (Bartolucci et al., 2023); we dedicate part of our appendix to a more formal definition of

this constraint (subsection A.1, Equation 36). This is the approach favored by *neural operator* methods Li et al. (2021); Kovachki et al. (2023); Fanaskov & Oseledets (2023). This approach provides the ability to vary sampling density *and* sampling window *independently*. This approach comes with an important drawback however: *compatibility with mainstream layers* is lost as the conversion between operator forms cannot be achieved without substantial alterations to typical layers (Bartolucci et al., 2023), which presents a significant barrier to more widespread adoption. In contrast, our approach follows the general paradigm of neural operators, but it entirely absorbs the conversion constraints within the fixed structure of Laplacian residuals, which provides *compatibility with mainstream layers*; our approach also skips layers as it reduces its sampling density, which enhances *computational adaptivity*.

**Adaptive-resolution through variable sampling density with implicit neural representations.** We can derive architectures with variable sampling density by leveraging an alternate representation of functions. While we can *explicitly* represent the input or output of layers as functions $s(\,\cdot\,) : \mathbf{X} \to \mathbb{R}^f$ through a set of samples $\{(\mathbf{x}_i, s(\mathbf{x}_i)) | \mathbf{x}_i \in \mathbf{X}_n\}$ that is tied to a *discrete coordinate space* $\mathbf{X}_n \subset \mathbf{X}$, we can instead *implicitly* represent functions $s(\,\cdot\,)$ through a parameter $\theta$ that is attached to a neural representation $n(\,\cdot\,, \theta) \approx s(\,\cdot\,)$ which approximates the function over the entire *continuous coordinate space* $\mathbf{X}$. This later approach is at the core of *implicit neural representation* methods Park et al. (2019); Mescheder et al. (2019); Sitzmann et al. (2020); Mildenhall et al. (2021); Chen et al. (2021); Yang et al. (2021); Lee & Jin (2022); Xu et al. (2022). Some of these methods directly leverage this scheme to reconstruct and render partially observed volumetric data, image data and light field data with great success (Park et al., 2019; Mescheder et al., 2019; Sitzmann et al., 2020; Mildenhall et al., 2021). Some of these methods expand on this scheme by splitting the parameter $\theta$ into a fixed part $\vartheta$ that is shared across all data points, and a latent part $z$ that is specific to individual data points, which enables forming more complex and reusable representations (Chen et al., 2021; Lee & Jin, 2022; Yang et al., 2021). This approach faces limited usefulness in the context of classification tasks, segmentation tasks, and diffusion tasks. This shortfall comes from a lack of *compatibility with mainstream layers*, and more broadly from a set of challenges that arise when mapping between implicit representations $n(\,\cdot\,, \vartheta, z_l) \mapsto n(\,\cdot\,, \vartheta, z_{l+1})$ directly through the latent space $z_l \mapsto z_{l+1}$, as such maps cannot easily preserve the symmetries of signals. This even renders difficult the implementation of convolutional layers (Xu et al., 2022) since there is *a priori* no simple relationship between the latent embedding $z_l$ of a signal $n(\,\cdot\,, \vartheta, z_l)$ and the latent embedding $z_{l+1}$ of the same signal convolved against a kernel $k_l$; there is within $z_l \mapsto z_{l+1}$ a nonlinear constraint $n(\,\cdot\,, \vartheta, z_{l+1}) = n(\,\cdot\,, \vartheta, z_l) * k_l$ that must be satisfied *everywhere* over the continuous coordinate space $\mathbf{X}$. In contrast, our method uses the most ubiquitous form of signal representation, which straightforwardly enables *compatibility with mainstream layers*.

## 2 Background

In our overview of background material, we introduce Laplacian pyramids as a stepping stone for the formulation of Laplacian residuals. In addition, we provide a discussion of signals in subsection A.1 that introduces the notation and fundamental concepts behind this work in a way that aims for accessibility.

### 2.1 Laplacian pyramids

Laplacian pyramids (Burt & Adelson, 1987) closely relate to Laplacian residuals, and are often used in vision techniques to decompose signals across a range of resolutions.

Laplacian pyramids take some signal $s$, and produce a series of lower and lower bandwidth signals $p_1^{\text{low}}, \ldots, p_{m+1}^{\text{low}}$ by convolving against a sequence of smoothing kernels $\phi_1, \ldots, \phi_{m+1}$. Laplacian pyramids then generate difference signals $p_1^{\text{diff}}, \ldots, p_m^{\text{diff}}$ that isolate the part of the signal that was smoothed away at each step, which intuitively correspond to a certain level of detail of the original signal. The operations that compose a Laplacian pyramid can be captured by a base definition and two simple recursive definitions:

$$p_1^{\text{low}} = s * \phi_1 \qquad\qquad \in S_1 \qquad\qquad (1)$$

$$p_n^{\text{low}} = p_{n-1}^{\text{low}} * \phi_n \qquad\qquad \in S_n = \{s | s * \phi_n = s\} \qquad\qquad (2)$$

$$p_n^{\text{diff}} = p_n^{\text{low}} - p_{n+1}^{\text{low}} \qquad\qquad \in D_n = \{d | d * \phi_n - d * \phi_{n+1} = d\} \subset S_n \qquad\qquad (3)$$

In Figure 3, we summarize the recursive formulation of Laplacian pyramids in a three-block pyramid; this is intended to allow easy comparison with the Laplacian residuals we later illustrate in Figure 4.

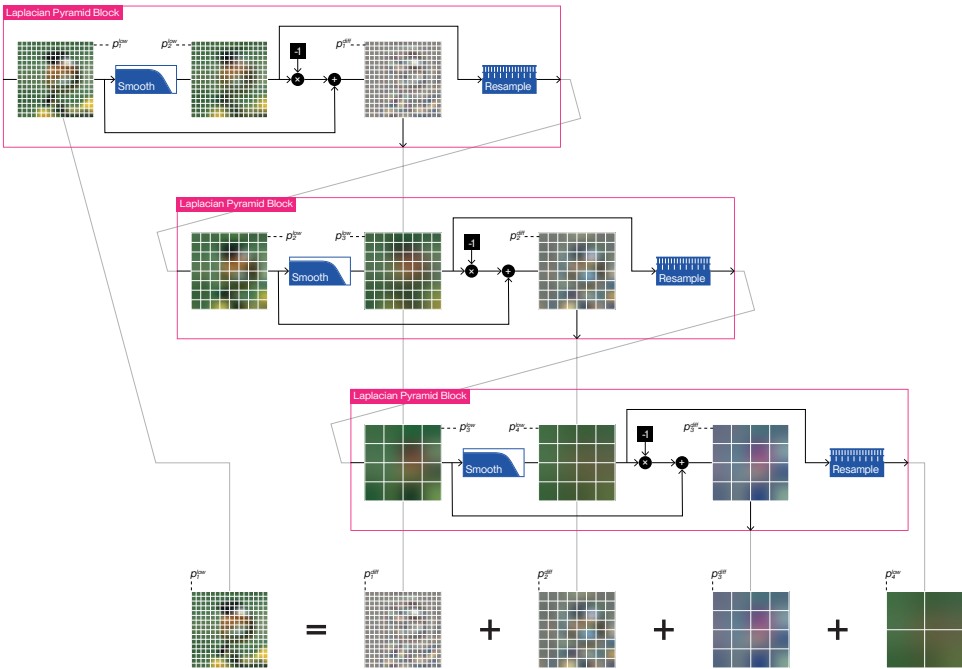

Figure 3: Visualization of three Laplacian pyramid blocks.

Laplacian pyramids are convenient to implement as each level reduces its computational cost relative to the preceding one. Each level fits within a stricter and stricter bandwidth constraint $S_1 \supset \cdots \supset S_m$. Each level therefore can be represented with a lower and lower sampling density $|\mathbf{X}_1|/\mathrm{area}(\mathbf{X}) > \cdots > |\mathbf{X}_m|/\mathrm{area}(\mathbf{X})$. We illustrate this in Figure 3. We represent signals in this way both in Laplacian pyramids and Laplacian residuals.

Laplacian pyramids enable reconstructing the original signal up to an arbitrary bandwidth $S_n$ using only the last lower bandwidth signal $p_{m+1}^{\mathrm{low}}$ and a variable number of difference signals $p_n^{\mathrm{diff}}, \ldots, p_m^{\mathrm{diff}}$. We illustrate this in the lower part of Figure 3. We express this as a linear decomposition more formally below:

$$\underbrace{p_n^{\mathrm{low}}}_{\forall \in S_n} = \underbrace{p_n^{\mathrm{diff}}}_{\exists! \in D_n} + \cdots + \underbrace{\underbrace{p_m^{\mathrm{diff}}}_{\exists! \in D_m} + \underbrace{p_{m+1}^{\mathrm{low}}}_{\exists! \in S_{m+1}}}_{\underbrace{\exists! \in S_m}} \qquad (4)$$
$$\underbrace{\phantom{p_n^{\mathrm{low}} = p_n^{\mathrm{diff}} + \cdots + p_m^{\mathrm{diff}} + p_{m+1}^{\mathrm{low}}}}_{\exists! \in S_n}$$

Laplacian pyramids offer *adaptive-resolution* with *variable sampling window*, since they rely entirely on convolutions that are equivariant to translation.

Laplacian pyramids also offer *adaptive-resolution* with *variable sampling density* through computation skipping. We can intuit this from Figure 3; if we were to start with an $8 \times 8$ image of the bird, it would appear reasonable to skip the $16 \times 16$ level of the Laplacian pyramid. We can deduce this is formally correct from Equation 4. We observe that any discrete signal $s[\ \cdot\ ]_u$ that is correctly sampled must respect the bandwidth constraint $S_u$ associated with its sampling pattern $\mathbf{X}_u$. We can therefore locate the bandwidth constraint of the signal relative to the bandwidth constraint of each level of the Laplacian pyramid — $S_1 \supset \cdots \supset S_n \supseteq S_u \supset \cdots \supset S_m$ — thus revealing the level of the Laplacian pyramid $p_n^{\mathrm{low}}$ that fully captures the signal. We know that all prior difference signals $p_1^{\mathrm{diff}}, \ldots, p_{n-1}^{\mathrm{diff}}$ must be zero in this case. We can ignore the computation of earlier levels of the Laplacian pyramid and start the computation immediately at $p_n^{\mathrm{diff}}$,

thus enabling *computational adaptivity*. We explicitly write the chain of zero terms that unravels in this scenario to later aid in understanding the case of Laplacian residuals, which is more complex:

$$
\begin{aligned}
p_1^{\text{low}} &= s * \phi_1 \\
&= s
\end{aligned}
\quad \Longrightarrow \quad
\begin{aligned}
p_2^{\text{low}} &= p_1^{\text{low}} * \phi_2 \\
&= s \\
p_1^{\text{diff}} &= p_1^{\text{low}} - p_2^{\text{low}} \\
&= 0
\end{aligned}
\quad \Longrightarrow \cdots \Longrightarrow \quad
\begin{aligned}
p_n^{\text{low}} &= p_{n-1}^{\text{low}} * \phi_n \\
&= s \\
p_{n-1}^{\text{diff}} &= p_{n-1}^{\text{low}} - p_n^{\text{low}} \\
&= 0
\end{aligned}
\tag{5}
$$

Laplacian pyramids are typically formulated using Gaussian smoothing kernels, which violate Equation 35 and introduce errors in the sampling process. We require *perfect* Whittaker-Shannon smoothing kernels for the computation skipping property we show in Laplacian pyramids above and in Laplacian residuals later in subsection 3.1 to hold exactly. We note that the form of decomposition yielded by Laplacian pyramids built using Shannon-Whittaker smoothing kernels is equivalent to the decompositions yielded by Shannon wavelets.

## 3 Method

In this section, we build towards *Laplacian residuals* (subsection 3.1), which are designed to allow the construction of *adaptive-resolution* architectures from *fixed-resolution* layers, and *Laplacian dropout* (subsection 3.2), which both serves as a regularizer for robustness at lower resolutions, and a regularizer for error-correction when imperfect smoothing kernels are used.

### 3.1 Laplacian residuals for adaptive-resolution deep learning

Laplacian residuals are alike to Laplacian pyramids in the way they decompose a signal into lower and lower bandwidth signals using smoothing filters, and in the way they are able to operate at lower resolution by simply skipping computations. However, Laplacian residuals crucially differ in their ability to incorporate neural architectural blocks that enable deep learning.

In Figure 4, we summarize the formulation of Laplacian residuals in a diagram that illustrates a chain of three Laplacian residuals, which allows easy comparison with the Laplacian pyramid shown in Figure 3.

Laplacian residuals are formulated as *adaptive-resolution* layers with *variable sampling density* $r_n : (\mathbf{X} \to \mathbb{R}^{f_n}) \in S_n \to (\mathbf{X} \to \mathbb{R}^{f_{n+1}}) \in S_{n+1}$ that incorporate neural architectural blocks $b_n : (\mathbf{X}_n \to \mathbb{R}^{f_n}) \to (\mathbf{X}_n \to \mathbb{R}^{f_n})$ that may be *fixed-resolution* layers or *adaptive-resolution* layers with a *variable sampling window*. In the later case, Laplacian residuals also inherit *adaptive-resolution* with a *variable sampling window*.

Laplacian residuals are compatible with any neural architectural block $b_n$ so long as it produces a constant everywhere when its input is zero everywhere (Equation 6), which is trivially the case for linear layers, activation layers, convolutional layers, batch normalization layers, some transformer layers, and for any composition of layers that individually meet this condition. This wide *compatibility with mainstream layers* is unseen in prior *adaptive-resolution* architectures with variable *sampling density*.

$$
b_n\{0\} = a \text{ where } a \in \mathbb{R}^{f_n} \tag{6}
$$

Each Laplacian residual block $r_n$ incorporates a single Laplacian pyramid block that is applied on the result of the preceding Laplacian residual $r_{n-1}$:

$$
p_n^{\text{low}}\{r_{n-1}\} = r_{n-1} * \phi_n \qquad\qquad \in S_n \tag{7}
$$
$$
p_n^{\text{diff}}\{r_{n-1}\} = p_n^{\text{low}}\{r_{n-1}\} - p_{n+1}^{\text{low}}\{r_{n-1}\} \qquad\qquad \in D_n \subset S_n \tag{8}
$$

The value of the zeroth Laplacian residual is taken to be a linear projection $\mathbf{A}_0 : \mathbb{R}^{f_1 \times f_0}$ that raises the feature dimensionality from $f_0 \in \mathbb{N}^+$ to $f_1 \in \mathbb{N}^+$ before the first neural architectural block:

$$
r_0 = \mathbf{A}_0 s \tag{9}
$$

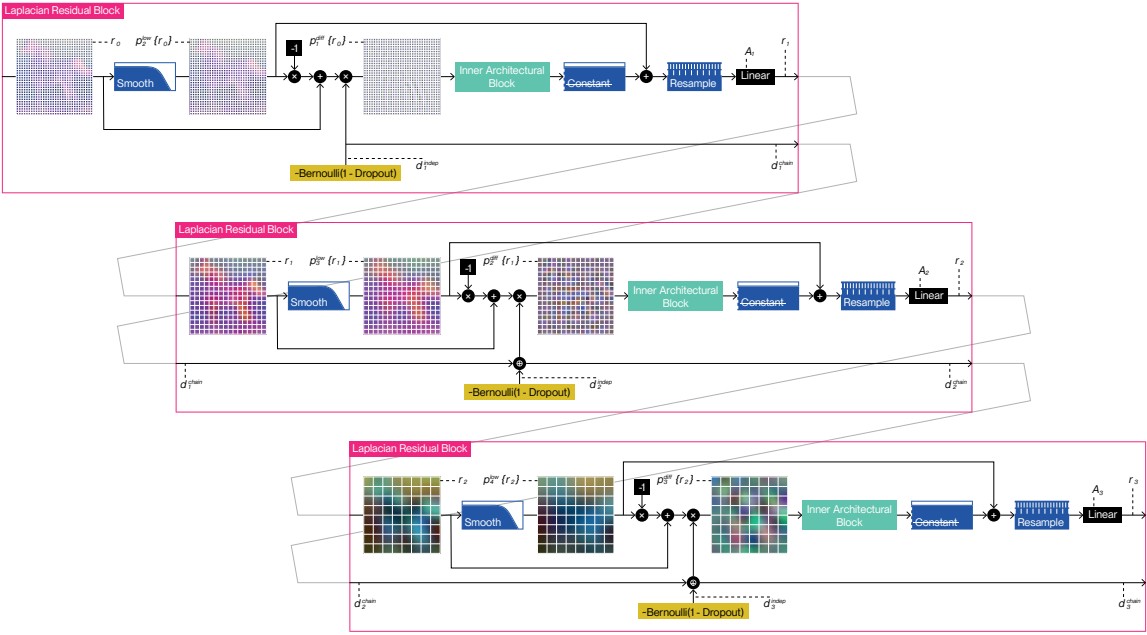

Figure 4: Visualization of a chain of three Laplacian residual blocks.

The value of each subsequent Laplacian residual is obtained by applying a neural architectural block $b_n$ on the difference signal $p_n^{\text{diff}}\{r_{n-1}\}$, by summing the lower bandwidth signal $p_{n+1}^{\text{low}}\{r_{n-1}\}$, and by applying some some additional processing, which we define below and motivate more concretely next:

$$r_n\{r_{n-1}\} = \mathbf{A}_n\left(b_n\{p_n^{\text{diff}}\{r_{n-1}\}\} * \psi * \phi_{n+1} + p_{n+1}^{\text{low}}\{r_{n-1}\}\right) \in S_{n+1} \tag{10}$$

We include a constant rejection kernel $\psi$ in Equation 10 to replicate the same computation skipping behaviour seen in Laplacian pyramids. This effectively subtracts the mean and ensures the neural architectural block $b_n$ contributes zero to the residual signal $r_n$ if the difference signal $p_n^{\text{diff}}\{r_{n-1}\}$ is zero, as shown in Equation 11. We obtain this result thanks to the constraint set on the neural architectural block $b_n$ in Equation 6:

$$b_n\{0\} * \psi = 0 \tag{11}$$

We include a smoothing kernel $\phi_{n+1}$ in Equation 10 to ensure the output bandwidth of $r_n$ coincides with the input bandwidth of $r_{n+1}$, which is necessary for Laplacian residuals to follow the same general structure as Laplacian pyramids.

We also incorporate a projection matrix $\mathbf{A}_n : \mathbb{R}^{f_{n+1} \times f_n}$ in Equation 10 to allow raising the feature dimensionality from $f_n \in \mathbb{N}^+$ to $f_{n+1} \in \mathbb{N}^+$ at the end of each Laplacian residual, so that more capacity can be allocated to later Laplacian residuals.

We note that the neural architectural block $b_n\{\cdot\}$ in Equation 10 is written in shorthand, and stands for the more terse expression $\mathfrak{I}_n\{b_n\{\mathfrak{S}_n\{\cdot\}\}\}$. This is a formal trick that enables our analysis by casting the neural architectural block from an *operator on discrete signals* $(\mathbf{X}_n \to \mathbb{R}^{f_n}) \to (\mathbf{X}_n \to \mathbb{R}^{f_n})$ to an *operator on continuous signals* $(\mathbf{X} \to \mathbb{R}^{f_n}) \in S_n \to (\mathbf{X} \to \mathbb{R}^{f_n}) \in S_n$ that is equivalent in the sense of Equation 36. This does not directly reflect the implementation of the method, as all linear operators are analytically composed then cast to their discrete form to maximize computational efficiency.

We add that we can alter the formulation of Equation 10 to let the neural architectural block perform a parameterized downsampling operation $(\mathbf{X}_n \to \mathbb{R}^{f_n}) \to (\mathbf{X}_{n+1} \to \mathbb{R}^{f_n})$ by changing the interpolation operator discussed above from $\mathfrak{I}_n : (\mathbf{X}_n \to \mathbb{R}^{f_n}) \to (\mathbf{X} \to \mathbb{R}^{f_n}) \in S_n$ to $\mathfrak{I}_{n+1} : (\mathbf{X}_{n+1} \to \mathbb{R}^{f_n}) \to (\mathbf{X} \to \mathbb{R}^{f_n}) \in S_{n+1}$ and by dropping the smoothing kernel $\phi_{n+1}$ from Equation 10.

**Adaptation to lower resolution signals with perfect smoothing kernels.** We guarantee that evaluating ARRNs while skipping Laplacian residuals is *exactly identical* to evaluating ARRNs while performing the interpolation step typically used to provide *adaptive-resolution* capability to *fixed-resolution* architectures. We specifically require the use of *perfect* smoothing kernels for this to hold. This guarantee provides strong theoretical backing to the validity of our method, and is also supported by empirical evidence in subsection 4.4.

We prove this property following the same argument leveraged to show *computational adaptivity* in the case of Laplacian pyramids. We are given a discrete signal $s[\,\cdot\,]_u$ that respects the bandwidth constraint $S_u$ implied by its sampling pattern $\mathbf{X}_u$. We know that this bandwidth constraint can be located relative to the bandwidth constraints of the Laplacian residuals — $S_1 \supset \cdots \supset S_n \supseteq S_u \supset \cdots \supset S_m$ — however we cannot immediately jump to claiming $p_1^{\text{diff}}, \ldots, p_{n-1}^{\text{diff}}$ must be zero because of the nonlinear effect of the inner architectural blocks $b_n, \ldots, b_{n-1}$. We must instead follow the terser process hinted at in Equation 5 with Laplacian pyramids and explicitly unroll the chain. We can see that the membership of $s$ to the bandwidth constraints $S_u \subseteq S_n \subset \cdots \subset S_1$ tells us $s$ is left unchanged by all of their smoothing filters. We can thus unroll the chain of Laplacian residuals while leveraging that $s = s * \phi_u = s * \phi_n = \cdots = s * \phi_1$ and that $b_n\{0\} * \psi = 0$ to arrive at our desired result:

$$r_0 = \mathbf{A}_0 s \tag{12}$$

$$p_1^{\text{low}}\{r_0\} = r_0 * \phi_1 \tag{13}$$

$$= \mathbf{A}_0 s \tag{14}$$

$$p_2^{\text{low}}\{r_0\} = r_0 * \phi_2 \tag{15}$$

$$= \mathbf{A}_0 s \tag{16}$$

$$p_1^{\text{diff}}\{r_0\} = p_1^{\text{low}}\{r_0\} - p_2^{\text{low}}\{r_0\} \tag{17}$$

$$= 0 \tag{18}$$

$$r_1\{r_0\} = \mathbf{A}_1 \left( b_1\{p_1^{\text{diff}}\{r_0\}\} * \psi * \phi_2 + p_2^{\text{low}}\{r_0\} \right) \tag{19}$$

$$= \mathbf{A}_1 \mathbf{A}_0 s \tag{20}$$

$$\vdots$$

$$p_n^{\text{low}}\{r_{n-2}\} = r_{n-2} * \phi_n \tag{21}$$

$$= \mathbf{A}_{n-2} \cdots \mathbf{A}_0 s \tag{22}$$

$$p_{n-1}^{\text{diff}}\{r_{n-2}\} = p_{n-1}^{\text{low}}\{r_{n-2}\} - p_n^{\text{low}}\{r_{n-2}\} \tag{23}$$

$$= 0 \tag{24}$$

$$r_{n-1}\{r_{n-2}\} = \mathbf{A}_{n-1} \left( b_{n-1}\{p_{n-1}^{\text{diff}}\{r_{n-2}\}\} * \psi * \phi_n + p_n^{\text{low}}\{r_{n-2}\} \right) \tag{25}$$

$$= \mathbf{A}_{n-1} \cdots \mathbf{A}_0 s \tag{26}$$

We can therefore evaluate a chain of Laplacian residuals at a lower sampling density by skipping higher sampling density Laplacian residuals $r_1, \ldots, r_{n-1}$ and instead starting the computation at $r_n$ while carrying over the linear projection $\mathbf{A}_{n-1} \cdots \mathbf{A}_0$. This provides *adaptive-resolution* with *variable sampling density* through *computational adaptivity* — without compromise in *robustness* — and without compromise in *compatibility with mainstream layers*.

We can use this result to precisely state the equivalence between evaluation using all Laplacian residuals after interpolation (Equation 27) and evaluation using the strictly necessary Laplacian residuals (Equation 28):

$$\mathfrak{S}_m\{r_m\{\ \cdots\ r_0\{\mathfrak{S}_0\{\mathfrak{I}_u\{s[\,\cdot\,]_u\}\}\}\ \cdots\ \}\} \tag{27}$$

$$= \mathfrak{S}_m\{r_m\{\ \cdots\ r_n\{\mathbf{A}_{n-1} \cdots \mathbf{A}_0 \mathfrak{S}_n\{\mathfrak{I}_u\{s[\,\cdot\,]_u\}\}\}\ \cdots\ \}\} \tag{28}$$

**Adaptation to lower resolution signals with approximate smoothing kernels.** We show that using *approximate* smoothing kernels *causes some numerical perturbation* when skipping the computation of higher sampling density Laplacian residuals. This observation motivates the use of Laplacian dropout, a training augmentation we later introduce in subsection 3.2 that addresses this limitation while also improving robustness.

When using *approximate* smoothing kernels $\varphi_n \approx \phi_n$ , the guarantee we provide does not hold exactly. We consider the case case where $\phi_n$ would leave a signal $s$ unchanged, and note that $\varphi_n$ would disturb the signal $s$ by a small error signal $\epsilon_n$:

$$s \in S_n \implies s * \phi_n = s \implies s * \varphi_n = s + \underbrace{s * (\varphi_n - \phi_n)}_{\epsilon_n} \tag{29}$$

We note the error above would cascade through every intermediate zero term that leads to Equation 26, therefore discarding unnecessary Laplacian residuals (Equation 28) would not be exactly equivalent to retaining all Laplacian residuals (Equation 27). We observe this is not simply constrained to a linear effect, as $\epsilon_1$ will for instance affect $b_1$, which has nonlinear behavior.

### 3.2 Laplacian dropout for effective generalization

In this section, we introduce Laplacian dropout, a training augmentation that is specially tailored to improve the performance of our method at effectively no computational cost.

We formulate Laplacian dropout by following the intuition that Laplacian residuals can be randomly disabled during training to improve generalization. We only allow disabling consecutive Laplacian residuals (using the logical or operator) to ensure that Laplacian dropout does not cut intermediate information flow:

$$d_n^{\text{indep}} \sim B(1 - d_n^{\text{rate}}) \tag{30}$$

$$d_n^{\text{chain}} = d_n^{\text{indep}} \oplus d_{n-1}^{\text{chain}} \tag{31}$$

$$p_n^{\text{diff}}\{r_{n-1}\} = d_n^{\text{chain}}(p_n^{\text{low}}\{r_{n-1}\} - p_{n+1}^{\text{low}}\{r_{n-1}\}) \tag{32}$$

Next, we provide a theoretical interpretation that identifies two distinct purposes that Laplacian dropout fulfills in our method. We see this dual utility as a highly desirable feature of Laplacian dropout.

**Regularization of robustness at lower resolution.** Since Laplacian dropout truncates Laplacian residuals in the same way they are truncated when adapted to lower sampling densities, Laplacian dropout is identical to randomly lowering sampling density when using *perfect* smoothing kernels. This acts as a training augmentation that promotes robustness over a *distribution* of lower resolutions. We perform a set of classification tasks in subsection 4.1 that show this regularizing effect sometimes *doubling* accuracy over certain lower resolutions without adversely affecting accuracy at the highest resolutions.

**Regularization of errors introduced by approximate smoothing kernels.** Since Laplacian dropout truncates Laplacian residuals in the same way they are truncated when adapted to lower sampling densities, Laplacian dropout exactly replicates numerical errors produced by *approximate* smoothing kernels in Equation 29. This allows learning a form of error compensation that offsets the effect of approximate smoothing kernels. We demonstrate in subsection 4.5 that this allows the use of very coarsely approximated smoothing kernels that otherwise impart a significant performance penalty on our method.

## 4 Experiments

We present a set of experiments that demonstrate our method's *robustness* across resolutions, its *computational adaptivity*, and its *compatibility with mainstream layers*. We show (subsection 4.1) that our method is highly robust across diverse resolutions; (subsection 4.2) that adaptation provides our method with a significant computational advantage; (subsection 4.3) that our method can generalize across layer types in a way that exceeds the capabilities of prior *adaptive-resolution* architectures with variable *sampling density*;

(subsection 4.4) that our theoretical guarantee for adaptation to lower resolutions with *perfect* smoothing kernels holds empirically; and (subsection 4.5) that our theoretical interpretation of the dual regularization effect of Laplacian dropout coincides with the behaviour we observe empirically when isolating the effect of *approximate* smoothing kernels, which improves the robustness of our method through two distinct mechanisms.

**Experiment design.**   We compare models in terms of their robustness across resolutions, their computational scaling relative to resolution, and their ease of construction. We follow a typical use case where we train each model at a single resolution and then evaluate over a range of resolutions. We consider the fluctuation of accuracy and inference time over resolution as the metrics of interest for our discussion. We perform a set of classification tasks that require models to effectively leverage the information of low-resolution to medium-resolution images; **CIFAR10** ($32 \times 32$) (Krizhevsky et al., 2009), **CIFAR100** ($32 \times 32$) (Krizhevsky et al., 2009), **TinyImageNet** ($64 \times 64$) (Le & Yang, 2015) and **STL10** ($96 \times 96$) (Coates et al., 2011).

**Model design and selection.**   For most of our experiments (subsection 4.1, subsection 4.2, subsection 4.4 and subsection 4.5), we construct ARRNs by using layers that take inspiration from MobileNetV2 (Sandler et al., 2018) and EfficientNetV2 (Tan & Le, 2021); detailed design choices are documented in subsection A.3. For the experiment that investigates generalization across layer types (subsection 4.3), we construct ARRNs by transplanting layers that are found across a range of mainstream architectures that support *adaptive-resolution* with *variable sampling window* but without *variable sampling density*: **ResNet18**, **ResNet50**, **ResNet101** (11.1M-42.5M) (He et al., 2016), **WideResNet50V2**, **WideResNet101V2** (66.8M-124M) (Zagoruyko & Komodakis, 2016), **MobileNetV3Small**, **MobileNetV3Large** (1.52M-4.21M) (Howard et al., 2019). We splice the sequence of layers that composes each mainstream architecture at points where resolution changes occur and nest each resulting subsequence of layers in a Laplacian residual with matching resolution. We discard the first two Laplacian residuals for MobileNetV3, as the resolution of the tailing Laplacian residuals otherwise becomes very small. For our choice of baseline methods, we consider mainstream architectures that again support *adaptive-resolution* with *variable sampling window* but without *variable sampling density*. We show they compromise *robustness* across diverse resolutions, yet they have no substantial advantage in ease of implementation or *compatibility with mainstream layers* compared to our method. We include all mainstream architectures discussed previously in this comparison, along with **EfficientNetV2S**, **EfficientNetV2M**, **EfficientNetV2L** (20.2M-117.2M) (Tan & Le, 2021). For the experiments that validate our theoretical analysis (subsection 4.4 and subsection 4.5), we perform an ablation study over the quality of the smoothing kernel, the use of Laplacian dropout at training time, and the use of adaptation at inference time.

**Model training and evaluation.**   All models are trained for 100 epochs at the full dataset resolution with identical hyperparameters that are described in subsection A.3. All models are then evaluated at the full dataset resolution and at a range of lower resolutions that are generated by interpolation. We have showed in our introduction that the mainstream architectures we cover display very weak *robustness* when *evaluated directly* (meaning with *computational adaptivity*; Figure 1), and that they perform much more reliably when *evaluated after an interpolation step that ensures the sampling window is kept constant* (meaning without *computational adaptivity*; Figure 2). In the comparisons we make between baseline methods and our method (subsection 4.1, subsection 4.2 and subsection 4.3), we display baseline methods *evaluated with interpolation*, as it provides the fairest chance to compete against our method. In our appendix (subsection A.3), we show a complimentary comparison that displays baseline methods *evaluated directly*, which is most interesting in terms of inference time; our method displays more aggressive computational savings as resolution decreases. We underline that all baseline methods share the same set of training runs across these figures; their *parameters* are *exactly identical*; only their *mode of evaluation* changes. In terms of evaluating our method, we skip unnecessary Laplacian residuals unless performing an ablation over adaptation. We note that in principle, we should always *round up* the number of required Laplacian residuals for in-between resolutions; however, we sometimes achieve greater robustness if we *round down*. This is the case with TinyImageNet and STL10 in subsection 4.1 and subsection 4.2, and with TinyImageNet in subsection 4.3. This effect likely results from more consistent statistical properties encountered when only evaluating Laplacian residuals that have full access to the part of the signal they usually address.

## 4.1 Robustness and the effectiveness of Laplacian dropout

We demonstrate that our method allows for greater low-resolution *robustness* than mainstream methods without compromise in high-resolution performance. Figure 5 shows four ablations of our method corresponding to permutations of two sets: either with Laplacian dropout (red lines) or without Laplacian dropout (black lines); and either with adaptation (solid lines) or without adaptation (dashed lines). We see that with Laplacian dropout and with adaptation (full red lines), our method outperforms every baseline method across every resolution and every dataset. We also find that, in contrast, without Laplacian dropout (black lines), our method shows much weaker generalization across resolutions, clearly demonstrating Laplacian dropout is effective as a regularizer for robustness to diverse resolutions.

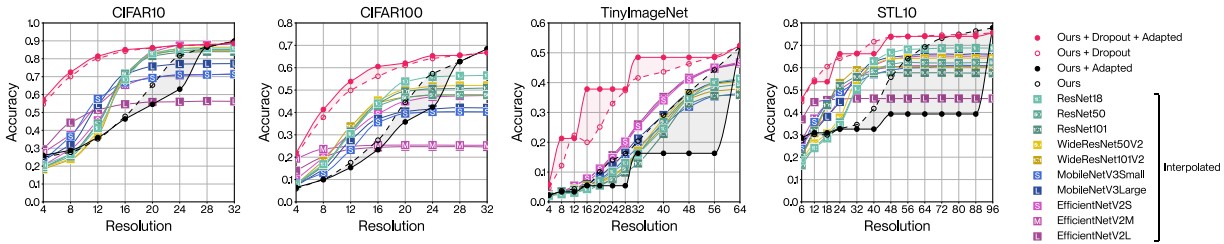

Figure 5: Accuracy of all architectures at various resolutions after training at the full dataset resolution. Evaluation is performed *after interpolation to the training resolution* in the case of mainstream methods. Our method (red full line) displays the best accuracy at the highest resolution and robustly maintains its accuracy at lower resolutions.

## 4.2 Computational efficiency

We confirm the advantage granted by *computational adaptivity* by performing inference time measurements on the previous experiment. We use CUDA event timers and CUDA synchronization barriers around the forward pass of the network to eliminate other sources of overhead, such as data loading, and sum these time increments over all batches of the full dataset. We repeat this process 10 times and pick the median to reduce the effect of outliers. Figure 6 shows the inference time of ARRNs with adaptation (full red lines) and without adaptation (dashed red lines). Our method significantly reduces its computational cost (highlighted by the shaded area) by skipping the evaluation of Laplacian residuals at lower resolutions. Our method also has a reasonable inference time relative to well-engineered mainstream methods.

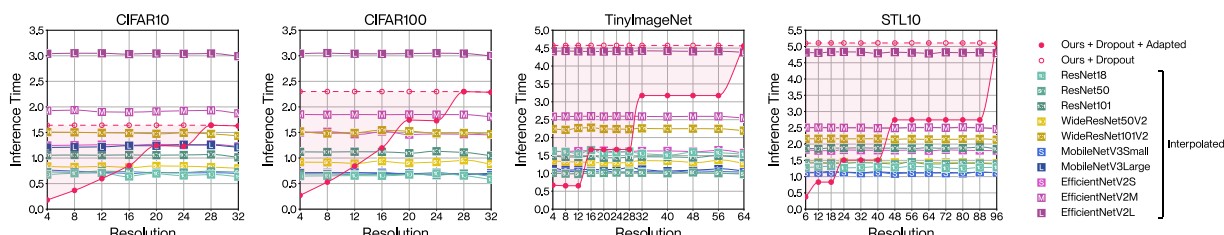

Figure 6: Inference time of all architectures at various resolutions. The inference time for the entire dataset is considered. Evaluation is performed *after interpolation to the training resolution* in the case of mainstream methods. Our method (red full line) can adapt to lower resolutions by skipping Laplacian residuals, which results in significant computational savings (highlighted by the shaded area) compared to using all Laplacian residuals (red dashed line). Our method also displays a reasonable inference time relative to typical convolutional neural networks despite not having a highly optimized implementation.

### 4.3 Generalization across layer types

We demonstrate the ease of use of our method and its *compatibility with mainstream layers* by constructing adaptive-resolution architectures from a range of mainstream architectures (**ResNet18**, **ResNet50**, **ResNet101**, **WideResNet50V2**, **WideResNet101V2**, **MobileNetV3Small** and **MobileNetV3Large**). Figure 7 compares the accuracy of architectures in adaptive-resolution form (red box plots) and in mainstream form (green box plots). The distribution of accuracies of the seven underlying architectures in the adaptive-resolution group and mainstream group is visually conveyed by drawing a small box plot at every resolution. Our method consistently delivers better low-resolution performance, and similar or better high-resolution performance. Our method achieves this while generalizing beyond the abilities of prior adaptive-resolution architectures with variable *sampling density*, which are incompatible with the layers used in this experiment.

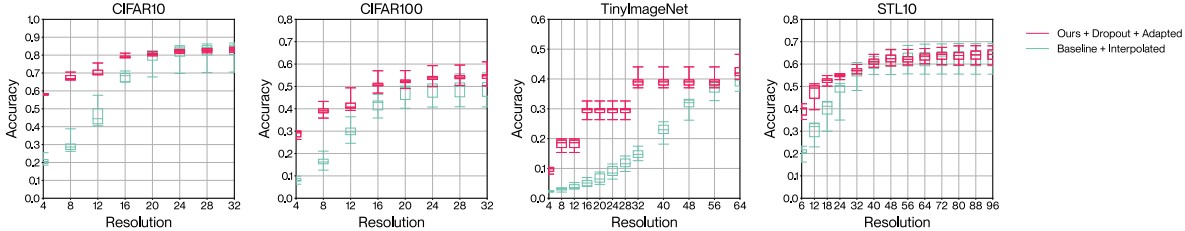

Figure 7:   Accuracy of two groups of methods at various resolutions, where 7 adaptive-resolution architectures (in red) are constructed by taking 7 mainstream architectures (in green) and wrapping their layers in Laplacian residuals. Evaluation is performed *after interpolation to the training resolution* in the case of mainstream architectures. Our method yields architectures that have stronger low-resolution performance, and similar or better high-resolution performance, which demonstrates ease of use and *compatibility with mainstream layers.*

### 4.4 Adaptation with perfect smoothing kernels

We perform an ablation study to verify our theoretical guarantee for numerically identical adaptation. Figure 8 displays a set of experiments that use *perfect quality* Whittaker-Shannon smoothing kernels (in the upper row of graphs in green) implemented through the Fast Fourier Transform (Cooley & Tukey, 1965). We showcase the usual set of ablations within this group of experiments; with Laplacian dropout (bright green lines) or without Laplacian dropout (dark green lines); with adaptation (full lines) or without adaptation (dashed lines). Our method evaluates practically identically whether unnecessary Laplacian residuals are discarded (with adaptation, full lines, Equation 28), or whether all Laplacian residuals are preserved (without adaptation, dashed lines, Equation 27). The discrepancies are either exactly zero, or are small enough to be attributed to the numerical limitations of floating point computation. The discrepancies are quantified thoroughly in the tables overlaid over each plot. Our method is thus able to skip computations without numerical compromise, as predicted by our theoretical guarantee.

### 4.5 Adaptation with approximate smoothing kernels and the dual effect of Laplacian dropout

We extend the previous ablation study to verify our theoretical analysis of the dual effect of Laplacian dropout. Figure 8 introduces a set of experiments that relies on *fair quality* approximate Whittaker-Shanon smoothing kernels (in the middle row of graphs in red), and on *poor quality* truncated Gaussian smoothing kernels (in the bottom row of graphs in blue). Figure 9 displays these same results in the form of a decision tree to help recognize the trends that are relevant to our discussion. This decision tree factors the impact of choosing a specific filter quality, choosing whether to use Laplacian dropout or not, and choosing whether to use adaptation or not. This analysis considers accuracy averaged over all resolutions and all datasets as the metric of interest. The numerical values displayed on each node correspond to the average multiplicative

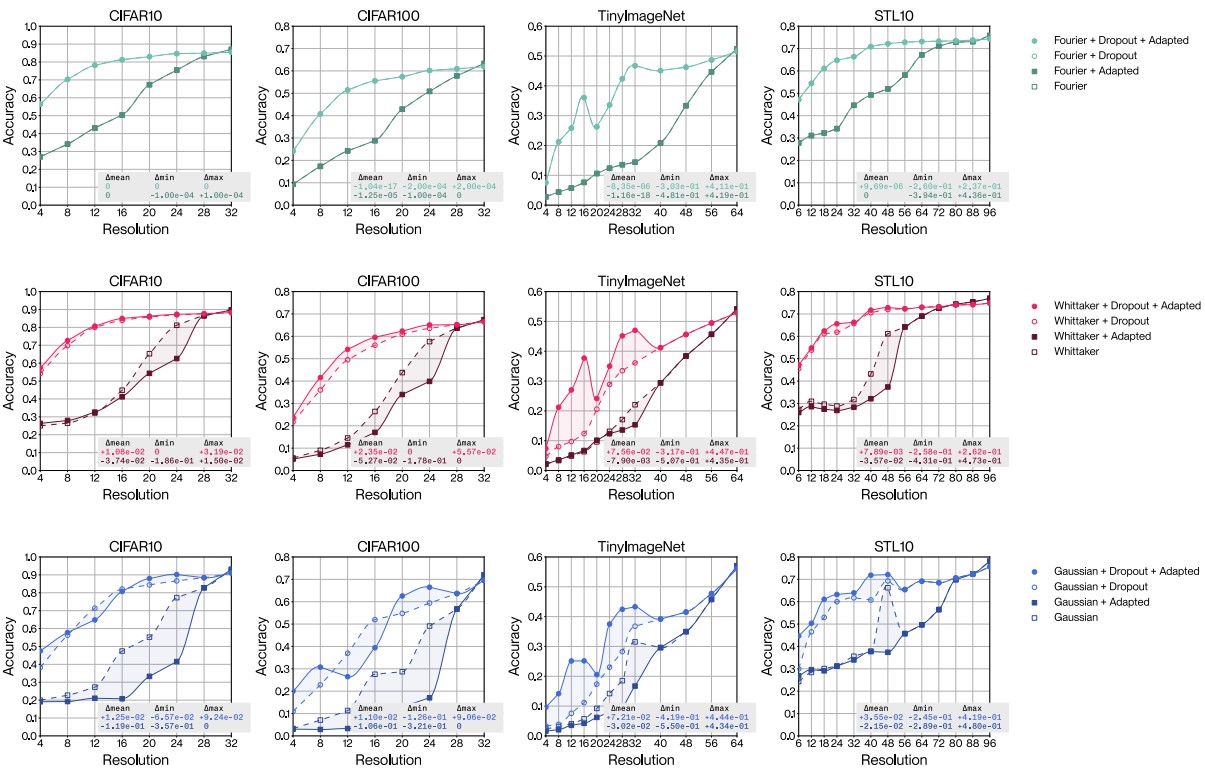

Figure 8: Accuracy of ARRNs at various resolutions, where each ARRN is identical to the architecture used in the main experiments aside from the choice of smoothing kernel. The smoothing kernels each correspond to a different row of graphs and a different hue. The shaded area spanning pairs of curves highlights the difference between the accuracy with adaptation and without adaptation. The overlaid tables display a statistical breakdown of this discrepancy with Laplacian dropout and without Laplacian dropout.

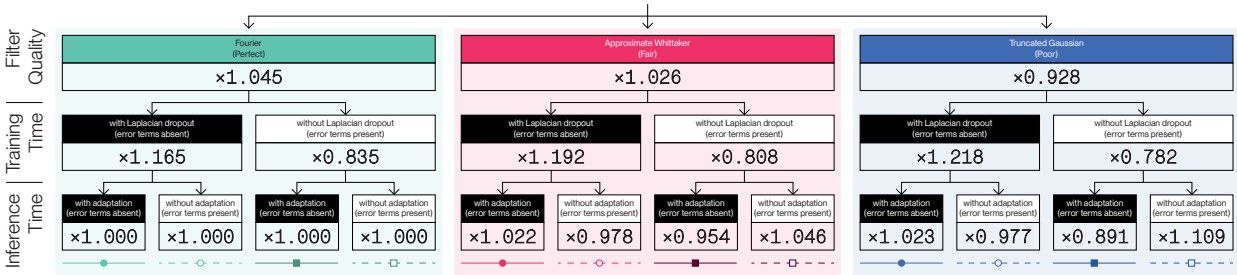

Figure 9: Multiplicative factor analysis showing an ablation over key components of ARRNs. The performance metric considered for analysis is the average accuracy over all resolutions and all datasets. The three levels of the decision tree show which filter is used, whether Laplacian dropout is used at training time, and whether adaptation is used at inference time. The numerical ratios indicate how average accuracy changes conditional to the decision associated with the node. The last two levels of nodes are coloured black or white according to the absence or presence of error terms $\epsilon_n$ respectively; at training time, this is determined by use of Laplacian dropout; at inference time, this is determined by use of adaptation. The symbols below the leaf nodes of the decision tree show the line style used in Figure 8 to allow easily referencing the underlying experiments in detail.

change in the metric once a decision is made. We claimed in subsection 3.2 that Laplacian dropout has two distinct purposes: it regularizes for robustness across a wider distribution of resolutions; it also mitigates numerical discrepancies caused by approximate smoothing kernels when using adaptation (Equation 29). We have demonstrated the first effect in subsection 4.1 and can also observe this effect clearly in the decision tree. We can only observe the second effect if we consider the choice of smoothing kernel, which motivates this experimental setup. Our method will conform to our theoretical interpretation if absence of error terms $\epsilon_n$ at training time yields better performance in the absence of error terms $\epsilon_n$ at inference time, and vice versa; that is to say on the last two levels of the decision tree, on the black and white nodes, tracing a path across two nodes of identical colour should yield a multiplier greater than 1 at the last node, and conversely, tracing a path across two nodes of opposing colour should yield a multiplier smaller than 1 at the last node. Our method displays exactly this behaviour, with the discrepancy at the last level of the decision tree growing monotonically with decreases in filter quality. Our method therefore leverages Laplacian dropout not just to improve robustness across a wider distribution of resolutions, but to compensate numerical errors induced by imperfect smoothing kernels, which enables the use of computationally greedy implementations.

## 5   Discussion

We have introduced ARRNs, a class of *adaptive-resolution* architectures that inherits the *compatibility with mainstream layers* of *fixed-resolution* methods, and the *computational adaptivity* and *robustness* of *adaptive-resolution* methods. ARRNs substitute standard residuals with *Laplacian residuals* which allow creating *adaptive-resolution* architectures using only *fixed-resolution* layers, and which allow skipping computations at lower resolutions without compromise in numerical accuracy. ARRNs also implement *Laplacian dropout*, which allows training models that perform robustly at a wide range of resolutions.

**Future Work.** We have provided evidence on classification tasks over low-resolution and medium-resolution image data; our method's ability to generalize is well supported by theoretical justification, but further experiments that include more challenging tasks and high-resolution data are desirable. We have investigated a form of Laplacian residual that *decreases* resolution, which addresses only a limited variety of architectures. We have applied our method in two dimensions on image data, but it is theoretically valid with any number of dimensions; its application to audio data and volumetric data is of interest.

**Impact Statement.** This paper presents work whose goal is to advance fundamental research in the field of deep learning and machine learning. No specific real-world application is concerned, although this contribution may render certain forms of technology more accessible.

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

# A    Appendix

## A.1    Background

We survey fundamental concepts of signal processing and introduce our notation. We aim to provide meaningful intuitions for readers who are not familiar with these principles, and to also rigorously ground our method and satisfy readers who are knowledgeable in this topic.

In Figure 10, we illustrate how the *sampling density* and *bandwidth* of signals intuitively relate to each other while also highlighting the notation and indexing scheme we use to designate these characteristics in our analysis. In Figure 11, we illustrate the effect of the *sampling window* over a signal. In the paragraphs that follow, we break down these notions in greater detail.

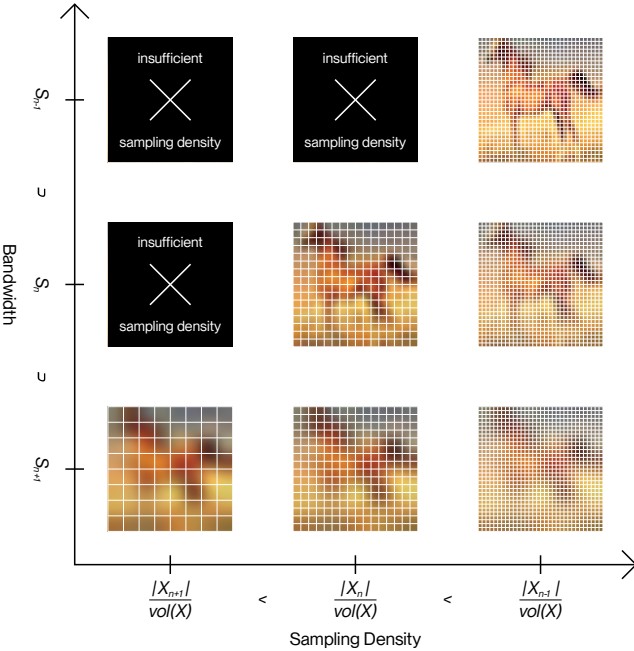

Figure 10:    Diagram showing the same signal captured with three distinct *sampling densities* $|\mathbf{X}_{n+1}|/\text{area}(\mathbf{X}) < |\mathbf{X}_n|/\text{area}(\mathbf{X}) < |\mathbf{X}_{n-1}|/\text{area}(\mathbf{X})$ (laid out horizontally) and three distinct *bandwidth constraints* $S_{n+1} \subset S_n \subset S_{n-1}$ (laid out vertically). The *sampling density* corresponds to the number of samples per unit of space used to cast the continuous signal into a discrete signal. The *bandwidth* corresponds to the smoothness of the underlying continuous signal. A continuous signal may only be captured into a discrete signal if sampling density is sufficient for the bandwidth; the blacked out signals show cases where this condition is not met.

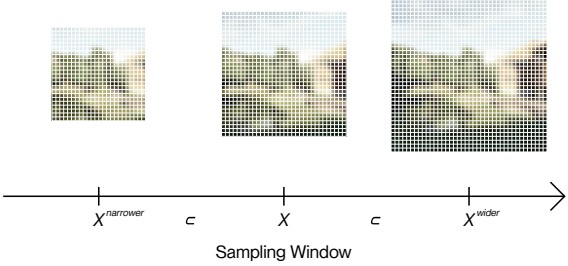

Figure 11:   Diagram showing the same signal captured with three distinct *sampling windows* $\mathbf{X}^{\text{narrower}} \subset \mathbf{X} \subset \mathbf{X}^{\text{wider}}$. The *sampling window* effectively consists in the region of space that is captured in the signal.

**Signals in continuous form.** We can represent signals as functions $s(\,\cdot\,) : \mathbf{X} \to \mathbb{R}^f$ that map a *continuous* spatial domain $\mathbf{X}$ that is a nicely behaved subset of $\mathbb{R}^d$ to a feature domain $\mathbb{R}^f$. This representation is useful because it allows us to leverage notions from functional analysis and calculus, and because it is fully independent from the way a signal is captured. We often refer to the *bandwidth* of signals in this work, which we can intuitively relate to the smoothness of continuous signals; low bandwidth signals are smooth, while high bandwidth signals are detailed.

**Signals in discrete form.** We can also represent signals as functions $s[\,\cdot\,]_n : \mathbf{X}_n \to \mathbb{R}^f$ that map from a *discrete* spatial domain $\mathbf{X}_n \subset \mathbf{X}$ to a feature domain $\mathbb{R}^f$. This representation enables us to perform computations on signals as they are fully defined by a finite amount of information given by the $|\mathbf{X}_n|$ individual samples $\mathbf{x}_i \in \mathbf{X}_n$. We commonly point to the quantity of samples as the *resolution* of a discrete signal. We associate indexing by $n$ to distinct resolutions throughout this work, where resolution *decreases* as $n$ *increases*, meaning $|X_n| > |X_{n+1}|$.

**Casting continuous signals into discrete signals by sampling.** We can easily take a continuous signal $s(\,\cdot\,)$ and create a discrete signal $s[\,\cdot\,]_n$ by sampling values at points $\mathbf{x}_i \in \mathbf{X}_n$. We notate this process as the *sampling operator* $\mathfrak{S}_n : (\mathbf{X} \to \mathbb{R}^f) \to (\mathbf{X}_n \to \mathbb{R}^f)$:

$$\mathfrak{S}_n\{s(\,\cdot\,)\} = \mathbf{x}_i \mapsto s(\mathbf{x}_i) \quad \forall \mathbf{x}_i \in \mathbf{X}_n \tag{33}$$

**Casting discrete signals into continuous signals by interpolation.** We can reverse the process above and derive a continuous signal $s(\,\cdot\,)$ from a discrete signal $s[\,\cdot\,]_n$ by applying a convolution with a smoothing kernel $\phi_n$, more formally known as a Whittaker-Shannon kernel (Whittaker, 1915; 1927). We note that sampling and interpolation are only inverses of each under certain important conditions we come back to later. We interpret the effect of the convolution against a smoothing kernel as filling in the gaps between the samples. We notate the process outlined here as the *interpolation operator* $\mathfrak{I}_n : (\mathbf{X}_n \to \mathbb{R}^f) \to (\mathbf{X} \to \mathbb{R}^f)$:

$$\mathfrak{I}_n\{s[\,\cdot\,]_n\} = \mathbf{x} \mapsto \sum_{\mathbf{x}_i \in \mathbf{X}_n} s[\mathbf{x}_i]_n \phi_n(\mathbf{x} - \mathbf{x}_i) \quad \forall \mathbf{x} \in \mathbf{X} \tag{34}$$

**Restricting the bandwidth of signals.** We need a slightly more formal way of designating signals that respect certain bandwidth constraints in order to better discuss the equivalence between discrete signals and continuous signals. We can use the smoothing kernels $\phi_n$ we just introduced to define sets of continuous signals $S_n = \{s | s * \phi_n = s\}$ that are already smooth enough to be left unchanged by the action of a smoothing kernel. We underline that convolving a smoothing kernel $\phi_n$ against a signal $s$ restricts its bandwidth such that it belongs to the corresponding set of signals $S_n$. We finally note that bandwidth constraints form an ordering $S_n \supset S_{n+1}$, meaning a signal $s$ that respects a low bandwidth constraint $S_n$ also respects any arbitrarily high bandwidth constraint $S_{n-k} \; \forall \; k > 0$.

**Equivalence of continuous signals and discrete signals.** We can use discrete signals or continuous signals to designate the same underlying information when the Nyquist-Shannon sampling theorem is satisfied (Shannon, 1949; Petersen & Middleton, 1962). This theorem intuitively states that a continuous signal with high *bandwidth* requires a discrete signal with correspondingly high *sampling density* for sampling to generally be feasible without error. This theorem more formally states that a discrete signal $s[\,\cdot\,]_n$ can uniquely represent any continuous signal $s(\,\cdot\,)$ that respects the bandwidth constraint encoded by membership to $S_n = \{s | s * \phi_n = s\}$, where the expression for the smoothing kernel $\phi_n$ depends on the *sampling density* $|\mathbf{X}_n|/\mathrm{area}(\mathbf{X})$ of the discrete spatial domain over the continuous spatial domain and on the assumptions on the boundary conditions of the continuous spatial domain $\mathbf{X}$ (Whittaker, 1915; 1927; Petersen & Middleton, 1962). We summarize the Nyquist-Shannon sampling theorem by stating that the sampling operator and interpolation operator are only guaranteed to be inverses of each other when the bandwidth constraint is satisfied:

$$s \in S_n \implies \mathfrak{I}_n\{\mathfrak{S}_n\{s\}\} = s \tag{35}$$

**Equivalence of operators acting upon continuous signals and discrete signals** We can extend the notion of equivalence between *continuous signals* $s(\,\cdot\,) : \mathbf{X} \to \mathbb{R}^f$ and *discrete signals* $s[\,\cdot\,]_n : \mathbf{X}_n \to \mathbb{R}^f$ to encompass the actions that can be performed on the same signals using *operators on continuous signals* $\mathfrak{O} : (\mathbf{X} \to \mathbb{R}^f) \to (\mathbf{X} \to \mathbb{R}^f)$ and *operators on discrete signals* $\mathfrak{O}_n : (\mathbf{X}_n \to \mathbb{R}^f) \to (\mathbf{X}_n \to \mathbb{R}^f)$. We are especially interested in this notion as it enables us to think of our neural architecture as a chain of operators that act on continuous signals that can be cast to act on discrete signals of any specific sampling density $|\mathbf{X}_n|/\text{area}(\mathbf{X})$. We often see this property formally labeled as *discretization invariance* in the neural operator community and highlight this concept is key to other works which allow adaptation to different sampling densities (Li et al., 2021; Kovachki et al., 2023; Fanaskov & Oseledets, 2023; Bartolucci et al., 2023). We can formally express the equivalence between the continuous form $\mathfrak{O}$ and discrete form $\mathfrak{O}_n$ of some operator as commutativity over the sampling operator when the bandwidth constraint is satisfied:

$$s \in S_n \implies \mathfrak{S}_n\{\mathfrak{O}\{s\}\} = \mathfrak{O}_n\{\mathfrak{S}_n\{s\}\} \tag{36}$$

## A.2 Related works

We shortly cover architectures that implement forms of residual connections that are similar to Laplacian residuals.

**Residual connections with filtering operations.** Singh et al. (2024) incorporates filtering operations within residuals to separate the frequency content of convolutional networks, although it provides no *adaptive-resolution* mechanism. Lai et al. (2017) uses Laplacian pyramids to solve super-resolution tasks with adaptive *output* resolution, with residuals ordered by *increasing* resolution. This is unlike our method, which is well suited to tasks with adaptive *input* resolution, with residuals ordered by *decreasing* resolution.

**Residual connections with dropout.** Huang et al. (2016) implements a form of dropout where the layers nested within residual blocks may be bypassed randomly. This is somewhat similar to Laplacian dropout, however, this is not equivalent to a form of bandwidth augmentation and does not result in the same improved robustness to various resolutions we show in subsection 4.1.

## A.3 Experiments

We include an alternate evaluation of the experiments we present in subsection 4.1 and subsection 4.2. We also provide further details on our experimental setup in this section.

In Figure 5 and Figure 6, we show baseline methods *evaluated after an interpolation step*, which yields stronger *robustness* but negates *computational adaptivity*. In Figure 12 and Figure 13, we instead show baselines *evaluated directly*, which yields weak robustness but provides some reduction in inference time at lower resolution. We observe that our method provides vastly superior robustness while also reducing its inference time more steeply at lower resolutions.

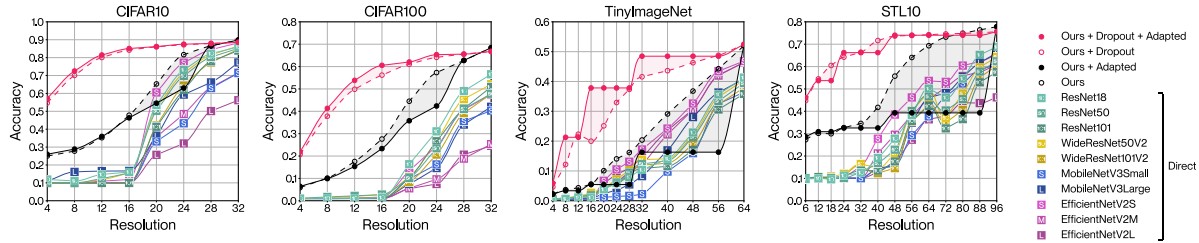

Figure 12: Accuracy of all architectures at various resolutions after training at the full dataset resolution. Evaluation is performed *directly* in the case of mainstream methods. Our method (red full line) more markedly dominates mainstream methods in terms of robustness under this alternate mode of evaluation.

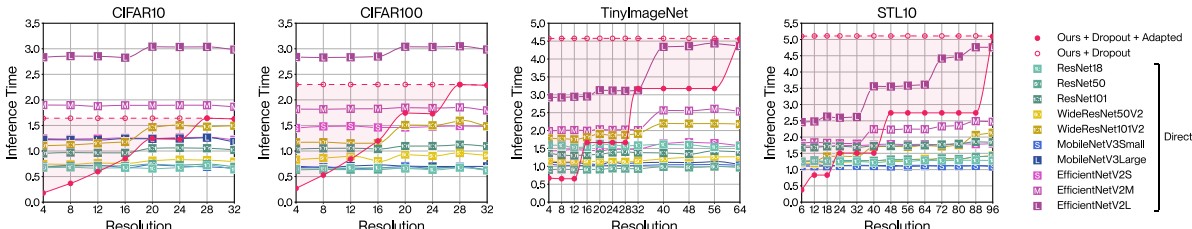

Figure 13: Inference time of all architectures at various resolutions. The inference time for the entire dataset is considered. Evaluation is performed *directly* in the case of mainstream methods. Our method (red full line) skips the computations of layers at lower resolutions, which more steeply reduces inference time relative to mainstream methods, which must still evaluate all their layers.

**Model design.** We provide detailed illustrations for the full architecture designs we use with our method according to each dataset: Figure 14 for CIFAR10 (5.33M-8.09M); Figure 15 for CIFAR100 (9.59M-14.5M); Figure 16 for TinyImageNet (15.0M-19.8M); and Figure 17 for STL10 (13.8M-18.4M). We indicate not a single parameter count, but a range of parameter counts for each architecture design, as adaptation enables computation of the forward pass or backward pass using a variable subset of the underlying parameters. We follow a general design pattern inspired by MobileNetV2 (Sandler et al., 2018) and EfficientNetV2 (Tan & Le, 2021) to derive these architecture designs. We nest inner architectural blocks ($b_n$ in Equation 10) within a series of Laplacian residual blocks of decreasing resolution and increasing feature count. We create these inner architectural blocks by composing depthwise $3 \times 3$ convolutions and pointwise $1 \times 1$ convolutions in alternation. We set all depthwise convolutions to use edge replication padding to satisfy Equation 6 and ensure resolution remains fixed within each Laplacian residual block. We prepend this string of layers with a pointwise convolution that expands the feature channel count. We conversely terminate the sequence of layers with a pointwise convolution that contracts the feature channel count inversely. We separate each convolution with a batch normalization (Ioffe & Szegedy, 2015) and a SiLU activation function (Elfwing et al., 2018), chosen for its tendency to produce fewer aliasing artifacts. We apply different Laplacian dropout rates ($d_n^{\text{rate}}$ in Equation 30) depending on the dataset: 0.6 for CIFAR10; 0.3 for CIFAR100; 0.3 for TinyImageNet; 0.3 for STL10. We use a common classification head that consists of a single linear layer with a dropout set to 0.2, which is applied after global average pooling. The designs were chosen by sweeping over different configurations for inner architectural blocks, and over different resolutions and number of features for the Laplacian residual blocks that contain them. The number of permutations per final sweep ranged between 18 to 216 for each dataset.

**Model training hyperparameters.** We provide the specific hyperparameters used during training.

For CIFAR10 and CIFAR100, across all methods, we use AdamW (Loshchilov & Hutter, 2019) with a learning rate of $10^{-3}$ and $(\beta_1, \beta_2) = (0.9, 0.999)$, cosine annealing (Loshchilov & Hutter, 2022) to a minimum learning rate of $10^{-5}$ in 100 epochs, weight decay of $10^{-3}$, and a batch size of 128. We use a basic data augmentation consisting of normalization, random horizontal flipping with $p = 0.5$, and randomized cropping that applies zero-padding by 4 along each edge to raise the resolution, then crops back to the original resolution.

For TinyImageNet and STL10, across all methods, we use SGD with a learning rate of $10^{-2}$, cosine annealing (Loshchilov & Hutter, 2022) to a minimum learning rate of 0 in 100 epochs, weight decay of $10^{-3}$, and a batch size of 128. We use TrivialAugmentWide (Müller & Hutter, 2021) to augment training.

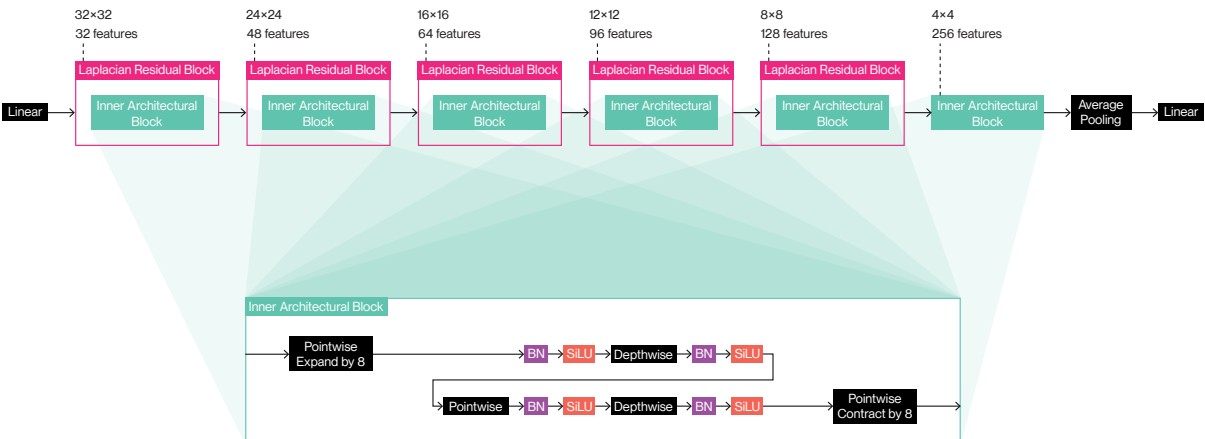

Figure 14: Architecture design for our method on CIFAR10.

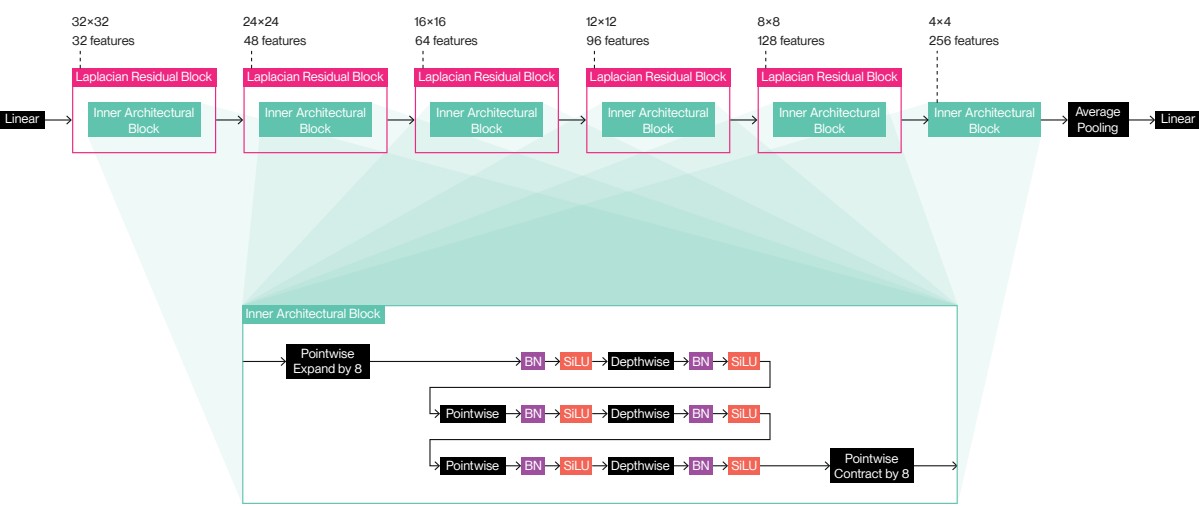

Figure 15: Architecture design for our method on CIFAR100.

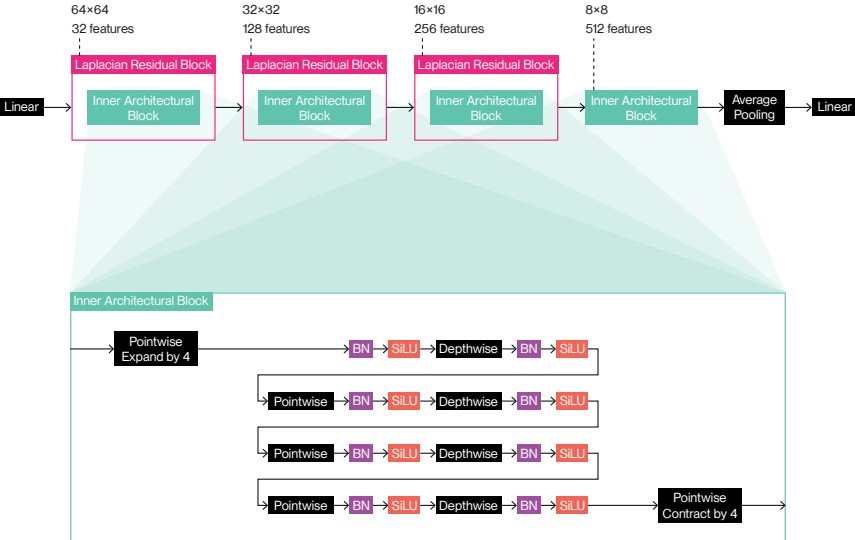

Figure 16: Architecture design for our method on TinyImageNet.

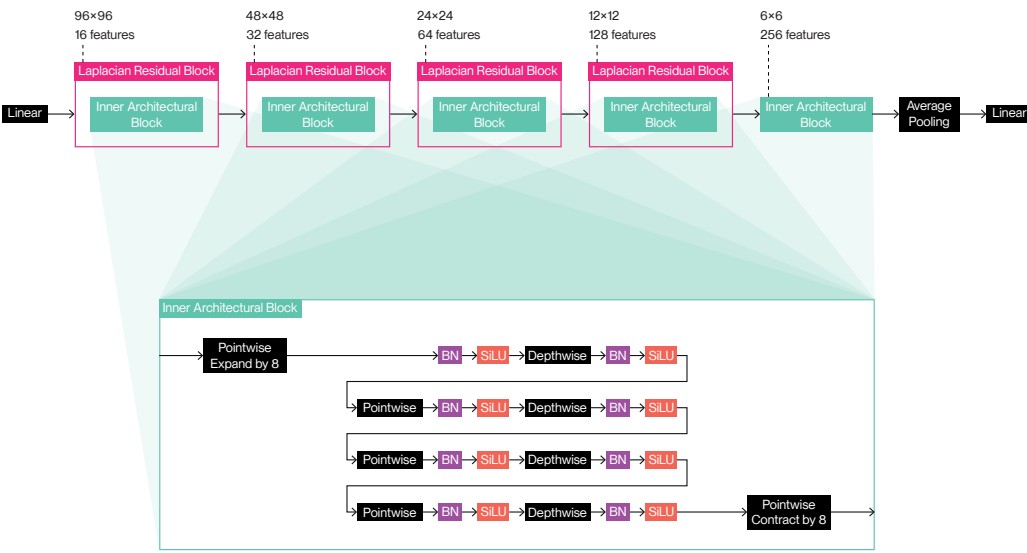

Figure 17: Architecture design for our method on STL10.

