# OpenReview forum: "Adaptive Resolution Residual Networks — Generalizing Across Resolutions Easily and Efficiently"
_TMLR — Accepted by TMLR_

### Review · Reviewer_xWy1 · 2025-02-10

**Summary Of Contributions:**

This paper presents a method that adapts common neural layers to various resolutions based on the Laplacian pyramid. The key idea is to build a Laplacian pyramid using low-pass filter, and make common neural layers operate on the residual signal. Experimental results show that the proposed method improves neural networks' generalization to downsampled inputs when trained on full resolutions.

**Audience:**

Yes

**Broader Impact Concerns:**

There is no ethical concern.

**Claims And Evidence:**

No

**Requested Changes:**

1. The authors should revise the introduction of this paper by giving an overview of the problem that this paper is solving, especially stressing the gap between existing research and the goal that is achieved in this study.
2. The authors should revise and improve the clarity of the paper, corresponding to the two points I mentioned in the **Weaknesses**.
3. The authors are encouraged to include a comparison to the baseline methods where the low-res inputs are not upsampled before sent to the network.

**Strengths And Weaknesses:**

# Strengths
1. The problem studied in this paper, i.e. making image processing neural networks easily adaptive to various resolutions, is important. Hence the technique developed by the authors and presented in this paper may have an impact on this area of research.
2. The authors have conducted extensive experimental evaluation to demonstrate the characteristics of the proposed method. The authors also included a lot of details in the paper, which may benefit follow-up researches.
3. The idea of adopting Laplacian pyramid for deep learning with variable resolutions is interesting.

# Weaknesses
1. Despite that the paper presented a lot of background and mathematical details for the proposed method, the motivation and key idea is still not clear to me. The paper mainly introduce the problem through an overview of the related work, but what is missing in the literature and where is the gap between the existing research and the goal is still unclear. From my point of view, the authors paid too much attention on the mathematical derivation of the method but neglected the introduction to the concept, idea, and philosophy, making the paper overall not so accessible to broader audience.
Specifically, the following are not so clear to me and are suggested to be revised:
* p.2 The authors motivate the proposed method by saying that the existing form of representation has shortcomings when mapping functions to functions. This statement needs further elaboration. What does mapping functions to functions refer to? Why is this important?
* p.6 The authors should define "numerical perturbation" in the context.

2. As described in the experimental settings, for the compared methods handling a lower resolution, the input is first interpolated to a higher resolution. I wonder whether this is the most appropriate treatment. To my knowledge, if the original architecture is like some convolutions followed by an adaptive global pooling, then there is no need to first upsample the input and process. The architecture itself should naturally supports different resolutions. Hence I think the comparisons conducted in the study might not be fair. The authors should at least include a setting where the low resolution inputs are directly fed into the baseline networks.

---

> ### Author Response · Authors · 2025-02-24
> **Revision implementing your feedback**
>
> Dear reviewer,
>
> We thank you for your engagement with our work, your constructive feedback, and your comments on the thoroughness of our experiments. We see this exchange as an opportunity to render our contribution clearer and more accessible. We have implemented all the changes you requested:
>
> 1. We have rewritten our introduction to provide a more accessible framing for our method. We focus on designing an efficient deep learning architecture that can handle data points with a computational investment that is proportional to difficulty. We consider data points that are signals, where our notion of difficulty is resolution. We provide a richer survey of prior attempts at this problem by underlining a distinction between two subtypes of *adaptive-resolution* methods. This distinction was only offhandedly addressed in A.2 in our original manuscript; it was not obvious to a wider audience and made the motivation for our method unclear; it also closely relates to the experiments you have requested. In short, changes in resolution can be achieved in two independent ways: by changing the *sampling window*, which corresponds to *the region of space that is observed*, and by changing the *sampling density*, which corresponds to *the number of samples packed into each unit of space*. We provide visual intuition for this distinction in Figure 10 and Figure 11\. We first discuss *adaptive-resolution* methods with *variable sampling window* (which include fully convolutional architectures and the baseline architectures used in our experiments); they reduce their computational cost at lower resolution *but* compromise robustness. We make clear the pragmatic impact of this shortcoming in Figure 1: when *evaluated directly* over natural images of various resolutions, these methods display unacceptable robustness. We observe very different behaviour in Figure 2: when *evaluated after interpolation*, they achieve much better robustness. We thereby motivate *adaptive-resolution* methods with *variable sampling density* (which include neural operator architectures and implicit neural architectures); they effectively support this form of interpolation at the level of their layers; they reduce their computational cost at lower resolution *without* compromising robustness *but* sacrifice compatibility with mainstream layers. We thus more clearly identify the gap our contribution addresses in the literature: we present an *adaptive-resolution* method that reduces its computational cost at lower resolution *without* compromising robustness *and without* sacrificing compatibility with mainstream layers.
>
> 2. We have revised the two areas of our paper that were lacking in clarity:
>
>     - We have edited our related works section to better outline the history of *adaptive-resolution* methods. In the first part, we introduce *neural operators* following a simple question: How can we adapt a convolutional layer to different *sampling densities*? We explain that functional analysis can help us interpolate the kernel from one *sampling density* to another. We then propose applying this way of thinking to all layers that compose an architecture, which leads to the framework of *neural operators*. In the second part, we explain *implicit neural representations* by investigating alternate ways of capturing functions. While functions can be captured by sampling their values at different points in space, functions can also be defined using a neural parameterization that implicitly assigns values over all of space at once. We incorporate additional notation so we can offer clearer definitions and more directly point to the challenges that occur when creating maps between several *implicit neural representations*. We believe these changes better convey the usefulness of thinking of inputs and outputs of layers as functions and better showcase the strengths and limitations of the methods we improve upon.
>
>     - We have added a short intuitive explanation of what numerical perturbations refer to in context.
>
> 3. We have included the comparison you requested. We show baseline methods *evaluated directly* and include plots for both accuracy and inference time in the appendix (Figure 12, Figure 13). We observe weaker robustness as expected. We nonetheless grasp meaningful insights from the inference time comparison: our method more aggressively reduces its computational cost as it lowers its resolution by skipping layers, whereas baseline methods must evaluate all their layers no matter the resolution. We persist in comparing baseline methods *evaluated after interpolation* in the experiments section (Figure 5, Figure 6\) as this provides baseline methods with a fairer chance to compete against our method.
>
> We thank you for participating in the review process. We are confident our response addresses your requests and improves the value of our contribution. We encourage you to revisit your evaluation of our work and look forward to your response.

---

### Review · Reviewer_ZLpD · 2025-03-01

**Summary Of Contributions:**

This paper presents Adaptive Resolution Residual Networks (ARRNs), a novel deep learning architecture that effectively generalizes across varying image resolutions. Unlike traditional deep learning models that operate on fixed-resolution inputs, ARRNs overcome this limitation by incorporating Laplacian residual and Laplacian dropout modules. The paper demonstrates better robustness against lower resolutions for classification tasks.

**Audience:**

Yes

**Claims And Evidence:**

Yes

**Requested Changes:**

1. What are the key differences between the Laplacian Pyramid and other multi-scale image analysis methods, such as the Wavelet transform? I suggest adding a few sentences in the Related Works section to clarify these differences and provide a comparative analysis.
2. How to choose the number of downsizing levels? How does the accuracy depends on n?
3. See point 4 in my comments above, adding experiment comparisons to other multi-scale models.

**Strengths And Weaknesses:**

Strength:
1. Dealing with various input resolutions is a practical challenge. Very interesting and important topic.
2. The proposed method demonstrates better robustness against lower resolution inputs.
3. Adaptation improves evaluation runtime-efficiency while Laplacian dropout improves accuracy and robustness.

My comments and questions:
1. It would be interesting to explore how the proposed architecture generalizes to higher resolutions. Instead of training the model on the highest resolution (e.g., 96x96 in STL10), the authors could consider training on lower resolutions and evaluating the model's performance on the original resolution. In addition, experiments with real datasets would be more interesting.
2. This paper demonstrates the effectiveness of the proposed methods primarily for classification tasks. How does the proposed method perform in regression tasks, such as image reconstruction? Although lower-resolution inputs contain less information, would classical neural networks still perform well given the expected lower-resolution reconstruction?
3. It appears that while the baseline methods perform poorly at very low resolutions, they are relatively stable within the higher resolution range. In comparison to the solid black line in Figure 5, the proposed architecture without Dropout and adaptation, which is more sensitive to resolutions different from the training resolution, could you explain why this happens?
4. How does the proposed method compare to other multi-scale architectures? While baseline models like ResNet and EfficientNet can handle different input sizes, they do not explicitly leverage multi-scale architectures. It is unclear whether it is the multi-scale architecture itself that promotes robustness to different resolutions.

---

> ### Author Response · Authors · 2025-03-22
> **Response to your questions and revision implementing your feedback**
>
> Dear reviewer,
>
> We thank you for your engagement with our work. We provide answers to your questions below and incorporate your feedback into our manuscript.
>
> **(RC1) What is the relationship between the Laplacian pyramid and other multi-scale image analysis methods, such as the Wavelet transform?** We have added a short statement highlighting that Laplacian pyramids constructed from Whittaker-Shannon smoothing kernels yield Shannon Wavelet decompositions. We appreciate your comment as it allows better connecting our contribution to multi-scale methods.
>
> **(RC3/Q4) What causal link is there between the multi-scale nature of the architecture and its robustness?** The robustness of our architecture is only partly explained by the use of *some* multi-scale structure. We provide theoretical and empirical insights that precisely isolate the determining factors to its robustness.
>
> On the theoretical side, our method is designed to satisfy a constraint (Subsection 3.1, Equation 36\) that ensures computations can be skipped at lower resolutions without compromising numerical exactness. The use of *some* multi-scale structure is not sufficient to satisfy this constraint. The use of the *specific* multi-scale structure we propose is necessary. Namely, we must use *perfect* Whittaker-Shannon smoothing kernels, and we must perform mean subtraction after the layers nested within each Laplacian residual.
>
> On the empirical side, we show that our theoretical guarantee for computation skipping without numerical degradation holds when using *perfect* smoothing kernels (Subsection 4.4); we show that Laplacian dropout has a strong causal effect on robustness (Subsection 4.1/4.4/4.5); we finally show that mainstream methods can have their pooling layers substituted by Laplacian residuals to benefit from the advantages of our method when using Laplacian dropout (Subsection 4.3), thereby marginalizing over the specific choice of layers nested within Laplacian residuals.
>
> The robustness of our architecture therefore derives from the *specific* multi-scale structure induced by Laplacian residuals *combined with* the use of Laplacian dropout.
>
> **(Q3) What explains the more brittle response of certain architectures (Figure 5, black full line) to different resolutions when adaptation is used *but* Laplacian dropout is *not* used?** The architecture displays brittleness in this scenario as it relies on *approximate* smoothing kernels, which doubly rely on Laplacian dropout for robustness when performing computation skipping. We motivate this interpretation both theoretically and empirically.
>
> On the theoretical side, we show error terms are induced when performing computation skipping with *approximate* smoothing kernels (Subsection 3.1, Equation 37); we motivate (Subsection 3.2) the latter of the two roles of Laplacian dropout from this observation — firstly it allows training our method over a variety of lower resolutions at once, which improves robustness whether using *perfect* smoothing kernels or *approximate* smoothing kernels — secondly it allows training our method while reproducing exactly the error terms caused by computation skipping when using *approximate* smoothing kernels, which encourages a form of error correction.
>
> On the empirical side, we validate this theoretical interpretation with an ablation study that disentangles the two roles of Laplacian dropout (Subsection 4.4/4.5).
>
> The architecture you have pointed at (black full line) is stripped of the two mechanisms it relies on for robustness, and performs poorly as a result. The architecture however performs very favourably when Laplacian dropout is enabled (red full line), as intended in the typical use case of our method.
>
> **(Q1) What generalization may we observe when the inference resolution is higher than the training resolution?** The architecture would generalize in a way that minimizes prior assumptions, and in a way that exactly coincides with inference resolution at the training resolution. We can see this must be the case by studying the way our method adapts its resolution. If a *lower* resolution is used, a *lower* number of Laplacian residuals need to be evaluated. If, conversely, a *higher* resolution is used, a *higher* number of Laplacian residuals would need to be evaluated. However, there are no additional Laplacian residuals which can be evaluated in this scenario, since the inference resolution is *higher* than the training resolution *by definition*. We must then interpolate the inference resolution to the training resolution (Equation 16). We therefore obtain results that are identical whenever the inference resolution is *higher than or equal to* the training resolution. This behaviour ensures predictable generalization. This behaviour also encodes the lack of a prior over the interpretation of finer features which were never seen during training.
>
> We thank you for participating in the review process and look forward to your response.

---

### Review · Reviewer_N159 · 2025-03-08

**Summary Of Contributions:**

This paper works on the problem of how to adapt the network structure to inputs of different resolutions. To this end, the authors propose the ARRN architecture, which is inspired by the mechanism of the laplacian pyramid of images. To adapt the idea of the laplacian pyramid to neural networks, especially CNNs, the authors propose the laplacian residual blocks which only applies non-linearity to the laplacian residuals. The authors also further propose laplacian dropout to skip randomly skil laplacian residuals during training for better generalization. To demonstrate the effectiveness of the proposed method, the authors conduct experiments on various dataset with very small resolution (CIFAR-10&100, TinyImagenet, STL10) with various baseline architectures (Resnet, MobileNet and EfficientNet), which demonstrates better performance on varying, especially low resolution inputs.

**Audience:**

Yes

**Broader Impact Concerns:**

The broader impact has been discussed in the paper.

**Claims And Evidence:**

No

**Requested Changes:**

Please refer to the weakness and try to make the following changes:
- Justify the claim about weaknesses of transformers and CNNs from a more practical perspective (wether the output really suffers from the claimed weaknesses) instead of only focusing on the model structure paradigm.
- Reorganize the paper and make the method description in the main paper more self-contained and clear.
- Discuss about the degradation case of the method, when the operation before the final inner block becomes linear.
- Using practical image resolution for all experiments.

**Strengths And Weaknesses:**

### **Strengths**
- The problem this paper addresses is interesting. Resolution adaptability is a practical but often overlooked issue, especially in computer vision applications. This problem is a special case of transformation equivariance, and this paper can inspire follow-up research on this topic.

- The Laplacian pyramid-inspired solution proposed in this paper is reasonable, leveraging conventional domain knowledge in convolutional neural networks. Additionally, the use of Whittaker-Shannon smoothing kernels is interesting and guarantees generalization across different resolutions.

- Except for the small image resolution, the experimental items are sufficient.


### **Weaknesses**
- The first major weakness of this paper is its claims regarding existing architectures:
  - "We find this approach to be unsatisfying, as it lacks computational adaptivity." However, this approach has proven very effective for current transformer-based architectures. For segmentation tasks, such as SAM, the input is resized to be compatible with the training resolution, yet the output achieves sufficiently good results after resizing back to the original resolution. Therefore, the authors should elaborate more on this claim and provide sufficient justification; otherwise, it contradicts established facts.
  - "In many practical applications, however, they offer weak generalization across resolutions..." Although convolution operations themselves are only translation equivariant, the learned feature representations can be somewhat equivariant to more complex transformations including scaling (applying to different resolutions), moderate rotations, and perspective transformations, especially for encoder-decoder architectures. This capability is fundamental for applications such as feature descriptor matching.

- The second major weakness is the organization and writing, especially Section 3:
   - Figure 4 could be illustrated more clearly.
   - The expanded equations 5-14 and 21-34 can be condensed; the current presentation is poorly structured.
   - Many descriptions are dense without sufficient explanations. For example, "This is a formal trick that enables our analysis by ....... Equation 19." The explanation appears to be in the appendix but would be better placed in the main paper.

- The third major weakness is the method design. Specifically, to adapt to different resolutions, the authors only apply nonlinearity to the Laplacian residuals within each Laplacian residual block, while the low-pass components undergo only linear transformations. In the lowest resolution case, inference degrades to only using the last "Inner Architectural Block." This effectively upgrades input RGB features directly to a much higher dimension (256/512) through linear transformation, contradicting best practices of progressively upgrading feature channels.

- The final major weakness of this paper is that all experiments are conducted on inputs with extremely small resolutions, only up to 96x96 pixels. Conclusions drawn from experiments at such low resolutions do not generalize well to practical cases. A significant potential reason is padding. In this paper, the authors adopt replication padding, which, although better than default zero-padding for low-resolution images, still interferes with convolution operations. In more practical higher-resolution scenarios, this interference is less severe.

In summary, although this paper addresses an interesting problem, its significance under practical settings is limited unless proven otherwise. Additionally, both the method design and experimental setup have fundamental issues requiring considerable effort to resolve.

---

> ### Author Response · Authors · 2025-03-22
> **Response to your comments and revision implementing your feedback**
>
> Dear reviewer,
>
> We thank you for engaging with our work. We provide a set of clarifications and a set of updates to our manuscript in response to your comments.
>
> **(RC1) Purpose.** We present a deep learning architecture that leverages *computational adaptivity* to suit signals of various resolutions while maintaining *robustness* and *compatibility with mainstream layers*. We address a gap in the literature by uniting these three desirable characteristics in a single architecture. We design this architecture around a theoretical guarantee which ensures *computational adaptivity* without compromise in numerical exactness (Section 3, Equation 36). We approach this guarantee with as much generality as possible: no assumptions are made on the dimensionality of the signals, on the resolution of the signals, on the number of Laplacian residual blocks; only minor assumptions are made on the layers embedded within Laplacian residual blocks. We arrive at a solution space covering a wide range of architectures. We validate our theoretical guarantee empirically (Subsection 4.4) and demonstrate the wide generalization over layer types granted by our minimal assumptions (Subsection 4.3).
>
> The strength of our contribution lies in abstracting away the specificities of a broad set of applications to address a fundamental problem that underlies them in a principled manner. The value of such an approach should not be dismissed given it provides a form of interpretability that is both highly desirable and arduously attainable in the field of deep learning research.
>
> **(RC4/W4) Experiment resolution.** We show our method compares favorably to existing methods, improves robustness (Subsection 4.1), and significantly reduces inference time at lower resolutions (Subsection 4.2) when tested over multiple datasets. We cover datasets spanning a range of resolutions similar to foundational work on neural operators (Kovachki et al., 2023\) published in JMLR. We fairly acknowledge the limitations of our experiments (Section 5). We note that no other reviewer has debated the accuracy of our claims in relation to the range of resolutions covered. We also note that experiments of higher resolution such as ImageNet would exceed the turnaround time of TMLR. We finally see no compelling evidence pointing to our method suddenly failing at high resolution given our theoretical guarantee is fully independent from this factor.
>
> **(RC1/W1.1) Claims on fixed-resolution methods.** As you have noted, fixed-resolution methods are satisfactory for a range of applications that present some variability in resolution. We are however interested in providing a solution that *fully leverages the information available at higher resolutions*, and that *incurs no waste of computation at lower resolutions*. We find fixed-resolution methods unsatisfactory in this sense. We have updated our discussion to avoid confusion.
>
> **(RC1/W1.2) Claims on adaptive-resolution methods with variable sampling window.** As you have argued, a form of *approximate invariance* to *sampling density* can be learnt by mainstream convolutional architectures. We stress that while they can *adapt* their *sampling window*, they cannot *adapt* their *sampling density*. The two are distinct, and the latter is the fundamental aim of our method as well as neural operator methods (Kovachki et al., 2023). We point to applications involving satellite imagery as an example where architectures must *adapt* their *sampling density* to match physical measurement units for consistent feature extraction. We have updated this part of our discussion to better convey the nuance you have pointed at.
>
> **(RC3/W3) Hypothesized weakness related to feature count.** The observation that our method presents a higher feature count at its first layers when adapted to a lower resolution is correct. The hypothesis that this consists in a weakness lacks theoretical or empirical grounding. We arrive at this assignment of feature count following the same intuition you point at — Laplacian residuals should increase their feature count progressively with depth — therefore truncating early Laplacian residuals results in a higher feature count at the first Laplacian residual encountered. We consistently show our method performs more robustly than mainstream methods at lower resolutions, even when all Laplacian residuals are skipped and feature count raises immediately to its maximum.
>
> **(RC2/W2.3) Structure.** The submitted version of the manuscript includes the complete mathematical background that underlies our method in the appendix. We find this conveys the main argument without overwhelming the reader with technical details. We are open to moving the content of the appendix to our background section if the action editor sees fit.
>
> We thank you for participating in the review process. We are confident our response addresses the comments you have formulated. We look forward to your response.

---

### Decision · Action_Editor_KcKK · 2025-05-10

**Recommendation:** Accept with minor revision

**Comment:**

The paper introduces the ARRN architecture, along with associated blocks and dropout, to enable networks to adapt to varying resolutions. Experimental results demonstrate enhanced performance. Post-discussion, two reviewers offer favorable feedback, while one presents a critical view. All reviewers find the proposed adaptive resolution approach intriguing and consider the Laplacian pyramid solution reasonable. However, the primary concern raised is that the empirical experiments predominantly rely on small datasets. The authors to are suggested incorporate additional experimental analyses and make more careful claims in the introduction.

**Audience:**

Yes

**Claims And Evidence:**

Yes

---

> ### Author Response · Authors · 2025-06-24
> **Camera-Ready Revision**
>
> Dear action editor,
>
> We are pleased to share the camera-ready version of our manuscript following your recommendation. We have incorporated the feedback of the reviewers to make for a clearer and more rigorous contribution. We have made significant improvements to the following sections:
>
> - **Introduction.** We make sure to better situate our method relative to existing methods following the feedback of reviewer xWy1. We distinguish between fixed-resolution methods, which lack *computational adaptivity*, and adaptive-resolution methods, which addresses this weakness. We separate adaptive-resolution methods into two subtypes that offer an opposite tradeoff: they either sacrifice *robustness* (methods with variable *sampling window*), or sacrifice *compatibility with mainstream layers* (methods with variable *sampling density*). We immediately support this distinction with motivating experimental results and include complimentary illustrations in our annex to facilitate understanding. We thus provide context that allows clearly identifying the novelty and usefulness of our contribution: it features *computational adaptivity*, *robustness*, and *compatibility with mainstream layers* all at once, unlike prior methods. We also make our claims more careful and precise with regards to existing methods following the input of reviewer N159.
>
> - **Related Works.** We improve the clarity of our related works section by better contextualizing the surveyed methods, and by building more gradually to the intuition that underlies them, as requested by reviewer xWy1.
>
> - **Background.** We highlight the link between Laplacian pyramids constructed using perfect smoothing kernels and Shannon wavelet decompositions following the suggestion of reviewer ZLpD.
>
> - **Method.** We improve the readability and completeness of our text following suggestions from reviewer N159.
>
> - **Experiments.** We supplement our experimental results on *robustness* (subsection 4.1) and *computational efficiency* (subsection 4.2) at the request of reviewer xWy1. We perform the same experiments using an alternative methodology whereby baseline methods are evaluated *directly at the inference resolution*, rather than *after interpolating to their training resolution*. We observe the alternative methodology yields much worse *robustness* while only weakly improving computational cost through *computational adaptivity*. We thus validate our initial methodological choice as it provides the baseline methods with the fairest chance to compete against our method. We further demonstrate the important distinction between adaptive-resolution methods with a variable *sampling window* — as with the baseline methods, which sacrifice either *robustness* or *computational adaptivity* depending on the methodology — and adaptive-resolution methods with a variable *sampling density* — as with our method, which is unique in its ability to offer *computational adaptivity*, *robustness*, and *compatibility with mainstream layers* all at once.
>
> We thank you and the reviewers for your engagement with our work.

---

> > ### Author Response · Authors · 2025-07-14
> > **Final Decision**
> >
> > Dear action editor,
> >
> > We look forward to your final decision regarding the acceptance of our work. Please let us know if any further action is needed on our part.
> >
> > Thank you again for your commitment to the review process.